# Explainable Concept Generation through Vision-Language Preference Learning for Understanding Neural Networks' Internal Representations

**Aditya Taparia** [1]   **Som Sagar** [1]   **Ransalu Senanayake** [1]

## Abstract

Understanding the inner representation of a neural network helps users improve models. Concept-based methods have become a popular choice for explaining deep neural networks post-hoc because, unlike most other explainable AI techniques, they can be used to test high-level visual "concepts" that are not directly related to feature attributes. For instance, the concept of "stripes" is important to classify an image as a zebra. Concept-based explanation methods, however, require practitioners to guess and manually collect multiple candidate concept image sets, making the process labor-intensive and prone to overlooking important concepts. Addressing this limitation, in this paper, we frame concept image set creation as an image generation problem. However, since naively using a standard generative model does not result in meaningful concepts, we devise a reinforcement learning-based preference optimization (RLPO) algorithm that fine-tunes a vision-language generative model from approximate textual descriptions of concepts. Through a series of experiments, we demonstrate our method's ability to efficiently and reliably articulate diverse concepts that are otherwise challenging to craft manually. Github: `https://github.com/aditya-taparia/RLPO`

## 1. Introduction

In an era where black box deep neural networks (DNNs) are becoming seemingly capable of performing complex tasks, our ability to understand their internal representations post-hoc has become even more important before deploying them in the real world. Among many use cases, such revelations help engineers in further improving models and regulatory bodies in assessing their correctness. To help these users, developing methods that communicate information in a human-centric way is essential for ensuring usefulness.

Humans utilize high-level concepts as a medium for providing and perceiving explanations. In this light, post-hoc concept-based explanation techniques, such as Testing with Concept Activation Vectors (TCAV) (Kim et al., 2018), have gained great popularity in recent years. Their ability to use abstractions that are not necessarily feature attributes or some pixels in test images helps with communicating these high-level concepts with humans. For instance, as demonstrated in TCAV, the concept of stripes is important to explain why an image is classified as a zebra, whereas the concept of spots is important to explain why an image is classified as a jaguar. Given 1) a set of such high-level concepts, represented as sample images (e.g., a collection of stripe and spot images) and 2) test images of the class (e.g., zebra images), TCAV assigns a score to each concept on how well the concept explains the class decision (i.e., zebra). Although concept-based explanations are a good representation, their requirement to create collections of candidate concept sets necessitate the human to know which concepts to test for. This is typically done by guessing what concepts might matter and manually extracting such candidate concept tests from existing datasets. While the stripe-zebra analogy is attractive as an example, where it is obvious that stripes is important to predict zebras, in most applications, we cannot guess what concepts to test for, limiting the usefulness of concept-based methods in testing real-world systems. Additionally, even if a human can guess a few concepts, it does not encompass most concepts a DNN has learned because the DNN was trained without any human intervention. Therefore, it is important to automatically find concepts that matter to the DNN's decision-making process.

As attempts to automatically discover and create such concept sets, several works has focused on segmenting the image and using these segments as potential concepts, either directly (Ghorbani et al., 2019) or through factor analysis (Fel et al., 2023; 2024). In such methods, which we refer to as retrieval methods, because the extracted concept set is

---

[1]School of Computing and Augmented Intelligence, Arizona State University, Tempe, United States of America. Correspondence to: Aditya Taparia <ataparia@asu.edu>.

*Proceedings of the 42$^{nd}$ International Conference on Machine Learning*, Vancouver, Canada. PMLR 267, 2025. Copyright 2025 by the author(s).

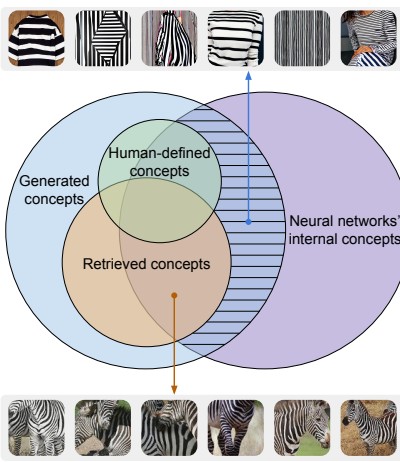

Figure 1: Humans can imagine (green) a few concepts to understand neural networks' representations (purple). Some other concepts can be retrieved from test images themselves through, for instance, segmentation (orange). However, if we generate concepts (blue), they will capture even a broader set of concepts. RLPO is designed to make this generation process more targeted toward neural networks' representations (purple ∩ blue).

already part of the test images as shown in Fig. 1 (Retrieved concepts), it is difficult for them to imagine concepts that do not have a direct pixel-level resemblance to the original image class. For instance, it is more likely that such methods provide patches of zebra as concepts instead of stripes.

By departing from existing concept set creation practices of human handcrafting and retrieval, we redefine concept set creation as a concept generation problem. Modern generative models such as stable diffusion (SD) can be used to generate realistic images. Nevertheless, since a generative model generates arbitrary images, we need to carefully guide the SD to generate explanatory images. One obvious approach is to engineer long, descriptive text prompts to generate concepts. However, engineering such prompts is not realistic. Therefore, to automate this process, as shown in Fig. 2, we propose a method, named reinforcement learning-based preference optimization (RLPO), to guide SD model. At its core, we devised a deep reinforcement learning algorithm, which gradually update SD weights to generate concept images that have a higher explanation score. The contributions of this paper can be summarized as follows:

1. We propose a method, named RLPO, to "generate" concepts that truly matter to the neural network (refer to the video on github). These include concepts that are challenging for humans or retrieval methods to anticipate (Fig. 1).

2. We qualitatively (computational metrics and human surveys) and quantitatively demonstrate that RLPO can reveal networks' internal representations.

3. Additionally, as use cases, we show how generated concepts can be utilized to improve models, understand model bias, and also demonstrate the generalizability of RLPO on tasks such as NLP sentiment analysis.

Notably, due to the modularity of our approach, any component in our framework (e.g., SD) can be seamlessly replaced with an improved version (e.g., SD3 (Esser et al., 2024) or Flash Diffusion (Chadebec et al., 2025)).

## 2. Preliminaries and Related Work

**Testing with Concept Activation Vectors (TCAV)**: The TCAV score quantifies the importance of a "concept" for a specific class in a DNN classifier (Kim et al., 2018). Here, a concept is defined broadly as a high-level, human-interpretable idea such as stripes, sad faces, etc. A concept (e.g., stripes), $c$, is represented by sample images, $X_c$ (e.g., images of stripes). In TCAV, a human has manually collected these sample concept images based on educated guesses, whereas our objective is to automatically generate them. For a given set of test images, $X_m$ (e.g., zebra images), that belong to the same decision class (e.g., zebra), $m$, TCAV is defined as the fraction of test images for which the model's prediction increases in the "direction of the concept." By decomposing the DNN under test as $f(x) = f_2(f_1(x))$, where $f_1(x)$ is the activation at layer $l$, TCAV score is computed as,

$$
\begin{aligned}
TS_{c,m} &= \frac{1}{|X_m|} \sum_{X_m} \mathbb{I}\left( \frac{\partial \text{output}}{\partial \text{activations}} \cdot (c \text{ direction}) > 0 \right) \\
&= \frac{1}{|X_m|} \sum_{x_i \in X_m} \mathbb{I}\left( \frac{\partial f(x_i)}{\partial f_1(x_i)} \cdot v > 0 \right)
\end{aligned}
\tag{1}
$$

Here, $\mathbb{I}$ is the indicator function that counts how often the directional derivative is positive. Concept activations vector (CAV), $v$, is the normal vector to the hyperplane that separates activations of concept images, $\{f_1(x); x \in X_c\}$, from activations of random images, $\{f_1(x); x \in X_r\}$. Refer to Appendix B.1 for details on the TCAV settings.

ACE (Ghorbani et al., 2019) introduced a way to automatically find concepts by extracting relevant concepts from the input class. It used segmentation over different resolution to get a pool of segments and then grouped them based on similarity to compute TCAV scores. Though the ACE concepts are human understandable, they are noisy because of the segmentation and clustering errors. As a different method, EAC (Sun et al., 2024) extracts concepts through segmentation. CRAFT (Fel et al., 2023) introduced a recursive strategy to detect and decompose concepts across layers. Lens (Fel et al., 2024) elegantly unified concept extraction and importance estimation as a dictionary learning

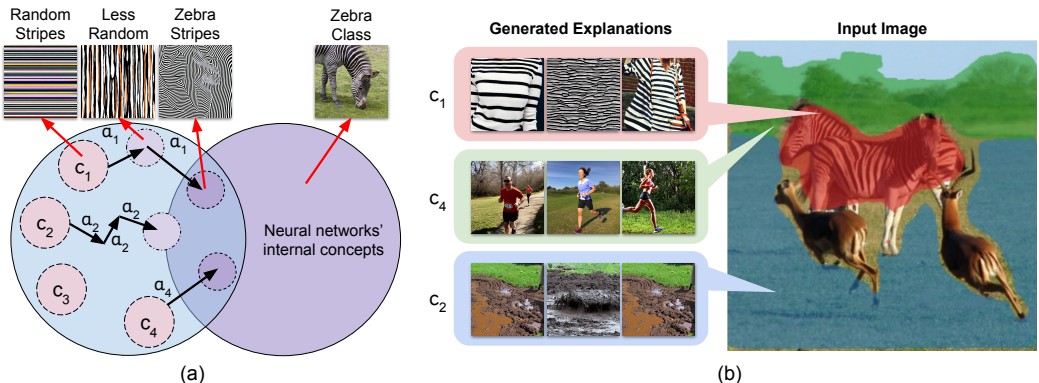

Figure 2: (a) Our proposed algorithm, RLPO, iteratively refines the concepts $c_i$ that are generated by a Stable Diffusion (SD) model by optimizing SD weights based on an action $a_i$. Each step in this update process provides an explanation at a different level of abstraction (Appendix C.4). Please refer to the supplementary video for better understanding. (b) Three concepts identified by our approach for the zebra class. Concepts are represented as images generated by SD.

problem. However, since all these methods obtain concepts from class images, the concepts they generate tend to be very similar to the actual class (e.g., a patch of zebra as a concept to explain the zebra, instead of stripes as a concept), making it challenging to maintain the "high-level abstractness" of concepts. In contrast, we generate concepts from a generative model.

**Deep Q Networks (DQN)**: DQN (Mnih et al., 2015) is a deep RL algorithm that combines Q-learning with deep neural networks. It is designed to learn optimal policies in environments with large state and action spaces by approximating the Q-value function using a neural network. A separate target network, $Q_{\text{target}}(s, a', \theta')$, Here $a'$ is $\arg\max Q(s_{next}, a)$ which is a copy of the Q-network with parameters $\theta'$, is updated less frequently to provide stable targets for Q-value updates,

$$
\begin{aligned}
Q(s_t, a_t) \leftarrow Q(s_t, a_t) + \alpha\Big(r(s_t, a_t) \\
+ \gamma \max_{a'} Q_{\text{target}}(s_{t+1}, a') - Q(s_t, a_t)\Big)
\end{aligned}
\tag{2}
$$

Here, $s_t$ is the state at step $t$, $a_t$ is the action taken, and $r_t$ is the reward obtained after taking action $a_t$. The parameters $\alpha$ and $\gamma$ are learning rate and discount factor, respectively. DQNs are used for controlling robots (Tang et al., 2025; Senanayake, 2024; Charpentier et al., 2022), detecting failures (Sagar et al., 2024), etc.

**Preference Optimization**: Optimizing generative models with preference data was first introduced in Direct Preference Optimization (DPO) (Rafailov et al., 2024). It is a technique used to ensure models, such as large language models, learn to align its outputs with human preference by asking a human which of its generated output is preferred. This technique was later extended to diffusion models in Diffusion-DPO (Wallace et al., 2024), where they updated Stable Diffusion XL model using Pick-a-Pic dataset (human preferred generated image dataset). Unlike traditional im-

age or text generation tasks, where the dataset for human preferred outputs are readily available, it is hard to have a general enough dataset for XAI tasks. To counter this problem, we provide preference information by using the TCAV score instead of a human, and use it to align the text-to-image generative model to generate concept images that matters for the neural network under test.

**Use of VLM in Explanation**: Recent advancements in Vision-Language models (VLMs) have open the doors for the use of VLMs in multiple domains, mainly because of their ability to generalize over large amount of data, they can be leveraged to obtain useful information. Work by (Sun et al., 2024) present a novel method combining the Segment Anything Model (SAM) with concept-based explanations, called Explain Any Concept (EAC). This method uses SAM for precise instance segmentation to automatically extract concept sets from images, then it employs a lightweight surrogate model to efficiently explain decision made by any neural network based on extracted concepts. Another work by Yan et al. (Yan et al., 2023) introduced Learning Concise and Descriptive Attributes (LCDA), which leverages Large Language Models (LLMs) to query a set of attributes describing each class and then use that information with vision-language models to classify images. They highlight in their paper that with a concise set of attributes, they can improve the classifier's performance and also increase interpretability and interactivity for end user.

## 3. Methodology: Reinforcement Learning-based Preference Optimization

Our objective is to find a set of concept images, $\mathcal{C}$, that maximize the TCAV scores, $TS_{c,m}$, indicating that the concepts are relevant to the neural networks' decision-making process. We leverage state-of-the-art text-to-image generative models to generate high-quality explainable concepts.

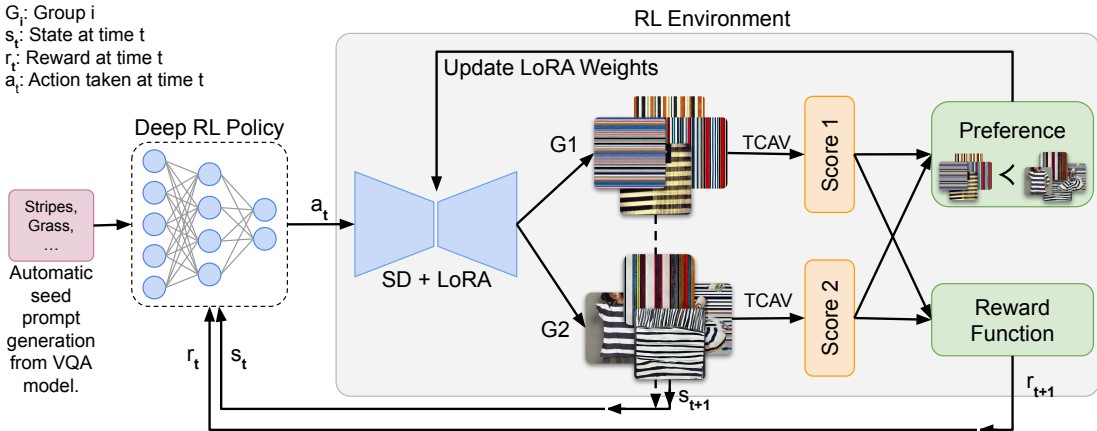

Figure 3: Overview of the RLPO framework with its dynamic environment interaction. The RL policy selects actions (seed prompts) which generates concept sets (G1, G2) scored through TCAV. Reward is calculated based on the scores obtained for both the sets. Simultaneously, best set is determined based on the scores obtained, which is used to update the LoRA layer of the SD model.

---

**Algorithm 1** The RLPO algorithm. Appendix B.2 for the expanded algorithm.

---

**Input:** Set of test images $f(\cdot)$
Run pre-processing and obtain seed prompts (action space)
**for** each episode **do**
    **for** each time step $t$ **do**
        Execute $a_t$ by picking a seed prompt
        Generate image groups $G_1$ and $G_2$
        Evaluate TCAV scores $TS_1$ and $TS_2$
        Update SD based on the better score
        Compute reward
    **end for**
**end for**
**Output:** Set of concept images

---

However, because the search space of potential text prompts is too large, we use deep RL to guide the image generation process. As described in Fig. 3 and Algorithm 1, our algorithm, RLPO, uses RL to pick potential keywords from a predefined list and iteratively optimizes stable diffusion weights to generate images that have a preference for higher TCAV scores. This process is described below.

**Notation**: Our framework contains three core deep learning models: the network under test $f(\cdot)$, the image generator $g(\cdot)$, and the deep RL network $h(\cdot)$. First, we have a pretrained neural network classifier that we want to explain. We then have a generative neural network, whose purpose is generating concept image sets, given some text prompts. In this paper, we use Stable Diffusion (SD) v1-5 as the generator as it is a state-of-the-art generative model that can generate realistic images. The core search algorithm that we train is a DQN.

## 3.1. The Rationales Behind Design Choices

We explain the design choices, which are validated through experiments in Section 4.1-4.4.

**Rationale 1: Why concept generation is a better idea.** If we use concept-based explanation the traditional way (Kim et al., 2018; Schut et al., 2025), then the end users need to manually guess what concepts to test for. Automatically retrieving the concept set by segmenting test images (Sun et al., 2024) also results in a limited concept set. In contrast, a SOTA generative model can generate high quality images. We provide more theoretical insights in Appendix A.

**Rationale 2: Why a deep RL-controlled VLM fine-tuning for generating concepts is a better idea.** *"A picture is worth a thousand words but words flow easier than paint."* As the saying goes, "a picture is worth a thousand words," it is much easier for people to explain and understand high-level concepts when images are used instead of language. For instance, we need a long textual description such as "The circles are centered around a common point, with alternating red and white colors creating a pattern" to describe a simple image of a dart board (i.e., Target Co. logo). Therefore, we keep our ultimate concept representation as images. However, controlling a generative model from visual inputs is much harder. Since human language can be used as a directed and easier way to seed our thought process, as the saying goes, "words flow easier than paint," we control the outcome by using text prompts. Since the vastness of the search space cannot be handled by most traditional search strategies, we resort to a DQN for controlling text. Since simple text alone cannot generate complex, high-level visual concepts, in each DQN update step, we use preference optimization to further guide the search process towards a more preferred outcome, allowing the DQN to focus on states

similar to the target. This approach improves our starting points for each DQN episode, enabling more efficient search and incremental progress towards the desired target.

### 3.2. Extracting Seed Prompts

Since a generative model can generate arbitrary images, if we provide good starting point for optimization then the convergence to explainable states would be faster. In this paper, to extract seed prompts for a particular class we make use of the off-the-shelf VQA model followed by several pre-processing steps, as described in Appendix B.3. We also explore how gibberish prompts can be used as seed prompts in Appendix B.4 which did not yield useful concepts.

### 3.3. Deep Reinforcement Learning Formulation

Our objective of using deep RL is automatically controlling text input of Stable Diffusion. As text input, we start with $\mathcal{K}$ seed prompts from Section 3.2, that have the potential to generate meaningful concept images after many deep RL episodes. We setup our RL state-action at iteration $t$ as,

- **Action** $a_t$: Selecting a seed prompt, $k_t \in \mathcal{K}$, that best influences concept image generation.
- **State** $s_t$: Preferred concept images generated from the seed prompt, $k_{t-1}$.
- **Reward** $r_t$: It is proportional to the TCAV score computed at state $s_t$ on action $a_t$, adjusted by a monotonically increasing scaling factor $\xi_{t,k}$. As each seed concept reaches the explainable state at different times, this factor is introduced to scale the reward over time $t$ for each unique seed concept $k$. Since the $g(.)$ is getting optimized at each time step $t$. The scaling factor is updated as $\xi_{t+1,k} \leftarrow \min\left(1, \frac{\xi_{t,k}+1}{T}\right)$, where T is total number of RL steps. Therefore, the expected cumulative adjusted reward is $R(\pi) = \mathbb{E}\left[\sum_{t=0}^{T} \xi_t \cdot r_t(s_t, a_t)\right]$.

Our objective in deep RL is to learn a policy, $\pi : s \rightarrow a$, that takes actions (i.e., picking a seed prompt) leading to explainable states (i.e., correct concept images) from proxy states (i.e., somewhat correct concept images). We formally define explainable state and proxy state as follow:

**Definition 1.** *Explainable states: States that have a concept score $TS_{c,m} \geq \eta$ for a user-defined threshold $\eta \in [0, 1]$ for concept $c$ and class $m$ is defined as an explainable state.*

**Definition 2.** *Proxy states: States that have a concept score $TS_{c,m} < \eta$ for the threshold $\eta \in [0, 1]$ for concept $c$ and class $m$ is defined as a proxy state.*

In practice, we set $\eta$ to a relatively large number, such as 0.7, to ensure that we look at highly meaningful concepts. In DQN, in relation to Eq. 2, we learn a policy that iteratively maximizes the $Q(s, a)$ value by using the update rule,

$$Q^*(s,a) = \mathbb{E}_{s' \sim P(\cdot|s,a)}[\xi_t r(s,a) + \gamma \max_{a' \in A} Q_{\text{target}}(s', a')]$$
(3)

### 3.4. Optimizing the States

At time $t$, the policy picks the seed prompt $k_t$, which is then used by the generative model, $g(k_t; w_t)$, with model weights $w$, to generate $2Z$ number of images. We randomly divide the generated images into two groups: $X_{c_1,t} = \{x_{c_1,t,i}\}_{i=1}^{Z}$ and $X_{c_2,t} = \{x_{c_2,t,i}\}_{i=1}^{Z}$. Let the TCAV scores of each group be $TS_{c_1,m,t}$ and $TS_{c_2,m,t}$. Since our objective is to find concepts that generate a higher TCAV score, concept images that have a higher score is preferred. Note that, unlike in the classical preference optimization setting with a human to rank, RLPO preference comes from the TCAV scores (e.g., $TX_{c_1,t} \succ TX_{c_2,t}$). We call this notion RLPO-XAIF in ablation studies below. If the generative model at time $t$ is not capable of generating concepts that are in an explainable state, $\max(TS_{c_1,m,t}, TS_{c_2,m,t}) \leq \eta$, we then perform preference update on SD's weights (more details in Appendix B.5). Following Low-Rank Adaptation (LoRA) (Hu et al., 2022)—a method that allows quick SD adaptation with a few samples,— we only learn auxiliary weights $a$ and $b$ at each time step, and update the weights as $w_{t+1} \leftarrow w_t + \lambda ab$.

As the deep RL agent progresses over time, the states become more relevant as it approaches explainable states (Fig. 2), thus the same action yields increasing rewards over time. To accommodate this, with reference to the rewards defined in Section 3.3, we introduce a parameter $\xi$, which starts at 0.1 and incrementally rises up to 1 as the preference threshold, $\eta$, is approached. Different actions may result in different explainable states, reflecting various high-level concepts inherent to $f(\cdot)$. Some actions might take longer to reach an explainable state. As the goal is to optimize all states to achieve a common target, DQN progressively improves action selection to expedite reaching these states. Thus, deep RL becomes relevant as it optimizes over time to choose the actions that are most likely to reach an explainable state more efficiently.

## 4. Experiments

To verify the effectiveness of our approach, we tested it across multiple models and several classes. We considered two CNN-based classifiers, GoogleNet (Szegedy et al., 2015) and InceptionV3 (Szegedy et al., 2016), and two transformer-based classifiers, ViT (Dosovitskiy et al., 2020) and Swin (Liu et al., 2021), pre-trained on ImageNet dataset. Unless said otherwise, only GoogleNet results are shown in the main paper. All other model details and results are provided in Appendix B.1 and C.3, respectively.

Table 1: Search strategy ablation. We see that RL, compared to $\epsilon$-greedy search, is the best strategy to explore the search space with high entropy, average normalized count (ANC) per action, and inverse coefficient of variance (ICV).

| Method | Entropy (↑) | ANC (↑) | ICV (↑) |
|---|---|---|---|
| RL (Ours) | **2.80** | **0.43** | **2.17** |
| 0.25 Greedy | 2.40 | 0.21 | 1.04 |
| 0.5 Greedy | 1.95 | 0.15 | 0.59 |
| 0.75 Greedy | 1.85 | 0.15 | 0.56 |

### 4.1. Ablation: Search Strategies (Why Deep RL?)

We chose DQN as our RL algorithm because of its ability to effectiveness traverse through discrete action space (Mnih et al., 2015) (20 unique seed prompts). We assess the effectiveness of RL by disabling the preference optimization step. As shown in Table 1, on the GoogleNet classifier, compared to $\epsilon$-greedy methods, the RL setup exhibits higher entropy, average normalized count (ANC), and inverse coefficient of variance (ICV) (See Appendix A.1 for definitions), indicating RL's ability to efficiently explore across diverse actions. Qualitative results obtained on updating SD model with and without RL are discussed in Appendix C.5.

### 4.2. Ablation: Scoring Mechanisms (Why TCAV?)

Another important aspect of our setup is the use of TCAV score, an XAI method, to provide preference feedback and calculate rewards. Alternatively, this XAI scoring feedback can also be replaced with human feedback or LLM-based AI feedback. As an ablation study, to test the effect of human feedback, we conducted human feedback experiment with eight human subjects who provided live human feedback. Further, to evaluate the LLM-based AI feedback, we made use of GPT-4o. More details on the experiment setup and results are provided in Appendix C.2. As shown in Table 2, we concluded that, even though other feedback techniques can be used, XAI-based feedback is best for generating concepts that are important to model with high speed and low computation cost. Though human and AI (GPT4o) are good at correlating semantics, by only looking at test images and concepts instead of model activations, they are not able to provide model specific explanations.

Table 2: Scoring mechanisms ablation. We see that RLPO with Explainable AI feedback (RLPO-XAIF), in this case TCAV, is a better choice than RLPO with human feedback (RLPO-HF) and AI feedback (RLPO-AIF).

| Method | Class-based Explanations | Model-specific Explanations | Feedback Cost* | Execution Time (↓) |
|---|---|---|---|---|
| RLPO-HF | ✓ | ✗ | NIL | $180 \pm 30s$ |
| RLPO-AIF | ✓ | ✗ | > 10 GB | $72 \pm 1.2s$ |
| RLPO-XAIF | ✓ | ✓ | < 1 GB | $56 \pm 0.7s$ |

\* Feedback cost refers to the GPU memory requirements.

Table 3: Exploration Gap (EG) and Odds for different methods based on the human survey (Appendix C.9). This verifies that RLPO can generate concepts that human cannot think of.

| | Laymen (n=260) | Expert (n=240) |
|---|---|---|
| EG (Retrieval) | 6.54% | 10.45% |
| EG (Ours) | 91.54% | 65.45% |
| Odds (Retrieval) | 14.29 | 8.57 |
| Odds (Ours) | 0.09 | 0.53 |

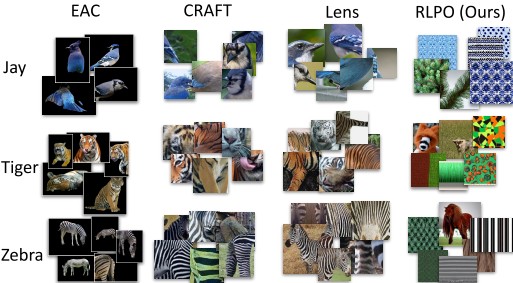

Figure 4: Samples of concepts generated by different methods. Observe that RLPO generates diverse images, beyond patches from the test images.

### 4.3. Concepts Generated by RLPO

Objective of RLPO is to automatically generate concepts that a human or a retrieval method cannot propose but the neural network has indeed learned (i.e., gets activated). As illustrated in Fig. 4, we observed that the RLPO can generate diverse set of concepts that a human would not typically think of but leads activations of the DNN to trigger. To validate this hypothesis, we conducted a survey to see if humans can think of these generated concepts as important for the DNN to understand a certain class.

As detailed in Appendix C.9, during our human survey, we presented a random class image followed by two concepts, one generated by our method and another from a previous retrieval-based method (Fel et al., 2024; 2023). While the choices were similar in terms of XAI-score from both methods, we discovered that most participants recognize retrieval-based concepts, and only those with domain-specific knowledge could identify generated concepts as important. As highlighted in Table 3, high Exploration Gap (EG) (definition in Appendix A.1) of our method indicates that most people can only identify concepts from a small subset of what $f(\cdot)$ learns during training. Intuitively, when we retrieve concepts from the test class, they tend to be similar to the test images. We further verify that the generated concepts have the following properties.

**Diverse representations per concept.** We verify the diversity (e.g., different types of stripes) of generated and retrieval-based concepts by computing the vector similarity of the CLIP and ResNET50 embeddings between $X_c$ and

Table 4: Novel concepts: $TS_{c,m}$ (TCAV score), CS (Cosine similarity), ED (Euclidean distance), RCS, and RED (CS and ED with ResNet50 embedding). This indicates that RLPO can generate a diverse set of concepts that triggers the network.

| Methods | Concepts | $TS_{c,m}(\uparrow)$ | CS ($\downarrow$) | ED ($\uparrow$) | RCS ($\downarrow$) | RED ($\uparrow$) |
|---|---|---|---|---|---|---|
| EAC (Sun et al., 2024) | C | 1.0 | $0.76 \pm 0.03$ | $7.21 \pm 0.63$ | $0.67 \pm 0.14$ | $6.34 \pm 2.16$ |
| Lens (Fel et al., 2024) | C1 | 1.0 | $0.77 \pm 0.02$ | $7.17 \pm 0.34$ | $0.50 \pm 0.18$ | $9.70 \pm 3.20$ |
| | C2 | 1.0 | $0.72 \pm 0.04$ | $8.02 \pm 0.87$ | $0.42 \pm 0.10$ | $10.90 \pm 2.80$ |
| | C3 | 1.0 | $0.69 \pm 0.05$ | $8.45 \pm 0.96$ | $0.45 \pm 0.05$ | $11.03 \pm 2.17$ |
| CRAFT (Fel et al., 2023) | C1 | 1.0 | $0.76 \pm 0.04$ | $7.37 \pm 0.62$ | $0.57 \pm 0.16$ | $8.80 \pm 3.20$ |
| | C2 | 1.0 | $0.72 \pm 0.02$ | $8.25 \pm 0.39$ | $0.50 \pm 1.90$ | $9.90 \pm 3.40$ |
| | C3 | 1.0 | $0.73 \pm 0.04$ | $7.98 \pm 0.79$ | $0.44 \pm 0.07$ | $10.80 \pm 1.90$ |
| RLPO (Ours) | C1 | 1.0 | $0.52 \pm 0.04$ | $10.48 \pm 0.50$ | $0.04 \pm 0.01$ | $16.80 \pm 1.40$ |
| | C2 | 1.0 | $0.49 \pm 0.02$ | $10.65 \pm 0.20$ | $0.02 \pm 0.02$ | $17.20 \pm 0.80$ |
| | C3 | 1.0 | $0.49 \pm 0.02$ | $10.74 \pm 0.30$ | $0.03 \pm 0.01$ | $17.60 \pm 4.40$ |

Table 5: Inter-concept Comparison for "zebra" class across three trial runs. Low Cosine Similarity (CS), high Wasserstein Distance (WD), and high Hotelling's T-squared Score (HTS) suggests that concepts generated from different seed prompts are different from one another.

| Metrics | Stripes-Running Concept | Running-Mud Concept | Mud-Stripes Concept |
|---|---|---|---|
| Avg. CS ($\downarrow$) | $0.67 \pm 0.01$ | $0.69 \pm 0.0004$ | $0.73 \pm 0.0004$ |
| Avg. WD ($\uparrow$) | $8.15 \pm 0.05$ | $7.85 \pm 0.02$ | $7.48 \pm 0.03$ |
| Avg. HTS ($\uparrow$) | $7598.50 \pm 84.5$ | $13069.68 \pm 2147.81$ | $7615.73 \pm 538.06$ |
| Are they from the same distribution? | No | No | No |

Table 6: Intra-concept Comparisons for "zebra" class across three trial runs. High Cosine Similarity (CS), low Wasserstein Distance (WD), and low Hotelling's T-squared Score (HTS) suggests that concepts generated from same seed prompts lies on the same distribution.

| Metrics | Stripes-Stripes Concept | Running-Running Concept | Mud-Mud Concept |
|---|---|---|---|
| Avg. CS ($\uparrow$) | $0.99 \pm 0.0008$ | $0.99 \pm 0.0004$ | $0.99 \pm 0.0004$ |
| Avg. WD ($\downarrow$) | $0.95 \pm 0.07$ | $0.82 \pm 0.07$ | $0.82 \pm 0.06$ |
| Avg. HTS ($\downarrow$) | $2.46 \pm 0.091$ | $2.36 \pm 0.08$ | $2.66 \pm 0.22$ |
| Are they from the same distribution? | Yes | Yes | Yes |

$X_m$ for different classes. As highlighted in Table 4, we observe that concepts from retrieval-based methods tend to have high cosine similarity with test images, making them less useful as abstract concepts (e.g., to explain the zebra class, a patch of zebra as a concept is less useful compared to stripes concept).

**Multiple concepts per class.** Since RLPO algorithm explores various explainable states, we can obtain multiples concepts (e.g., stripes, savanna) with varying level of importance. Fig. 5 shows the top three class-level concepts identified by our method for the "zebra" class for the GoogleNet classifier. We see that, each concept set has a different TCAV score associated with them indicating their importance. Additionally, as shown in Table 5, these concepts are inherently different from one another. As an additional result, we can see in Appendix C.4 how each concept evolves—representing different levels of concept abstractions—with each RL step.

### 4.4. Are Generated Concepts Correct?

After generating the concepts, next step is to identify what those concepts signify. To locate where in the class images generated concepts correspond, we made use of

CLIPSeg (Lüddecke & Ecker, 2022), a transformer-based segmentation model which takes in concept images as prompts, $X_c$, and highlights in a test image, $x \in X_m$, which part resembles the input prompt as a heat map. More details on this is available in Appendix C.3.3. As shown in Fig. 5, class image on left highlights the top 3 identified concepts by RLPO. We also compare the output generated by other popular XAI techniques such as LIME and GradCam with ones generated by RLPO, more details in Appendix C.6.

After finding the relationship between generated concepts and input images, we need to validate the importance of the identified concepts. To that end, we applied c-deletion, a commonly used validation method in XAI, to the class images for each identified concept. We gradually deleted concept segments based on the heat map obtained from ClipSeg. The results for the c-deletion are shown in the Fig. 6. We see the area under curve is the highest for the most important concept "stripes" and the lowest for least important concept "mud," indicating the order of importance of each concept. More examples on the c-deletion are in Appendix C.3.4. Additionally, to verify if the generated concepts are consistent, we compared their CLIP embedding across multiple runs. As shown in Table 6, we can see that

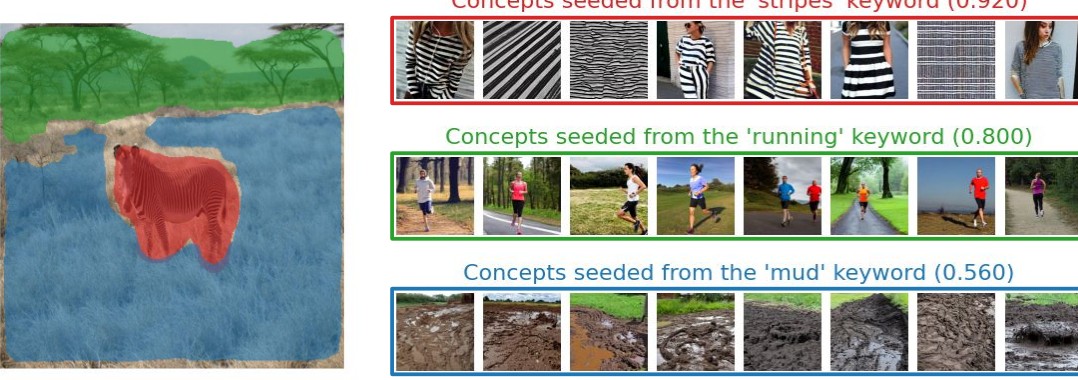

Figure 5: The figure shows the concepts generated by our method and where they are located in the input image ("zebra" class) for GoogleNet classifier. As highlighted the "stripes" concept images are located near zebra, the "running" concept images, showing trees, highlight the background, and the "mud" concept highlights the grass and soil in the input image. The concepts are ordered in their importance (TCAV score) with "stripes" being the highest and "mud" being the lowest.

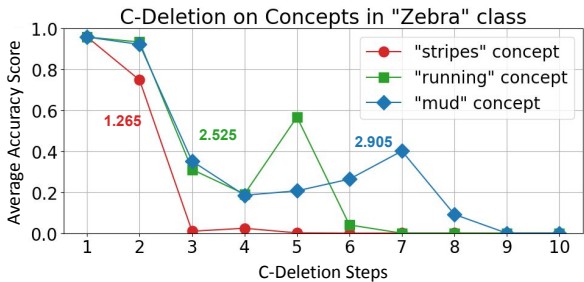

Figure 6: C-deletion. Removing concepts over time to measure the reliability. The colored numbers indicate the area under the curve.

the concepts generated from same seed prompt belongs to the same distribution across multiple runs.

### 4.5. How Are Generated Concepts Useful to Engineers?

To verify the usability of the generated concepts, we conducted a human study with 19 ML engineers. We first provided them the concept generated by our method for "zebra" class and ask them to choose relevant concepts for GoogleNet to classify a zebra without telling them that all shown images are actual concepts. All the engineers selected the "stripes" concept to be important while some also selected the "mud" concept. But most missed the "running" concept. This indicates that engineers cannot think of all the important concepts that gets the neural network activated. In the next step, we showed engineers the concept-explanation mapping on a random input image (similar to Fig. 5) and asked them if the provided explanation helped them understand the model better and if it provided new insights. 94.7% of the engineers agreed that the explanation helped in better understanding the neural network and 84.2% agreed that it provided new insights. This result shows that the new concepts discovered by our proposed method help engineers discover new patterns that they did not imagine before (More details in Appendix C.10).

We now demonstrate how engineers can use these newly revealed information about concepts to improve the model. As identified by RLPO, for Tiger class, the base GoogleNet model gives equal importance to both foreground (highlighted by concepts "orange black and white" and "orange and black") and background (highlighted by concepts "blurry") in the input (see Fig. 12 in Appendix C.3 for example explanations). As shown in Fig. 7, when we fine-tune the GoogleNet on images of Tiger-related concepts, we see that the fine-tuned model now focuses more on tiger than the background while maintaining a similar accuracy (65.6%). Details in Appendix C.8.

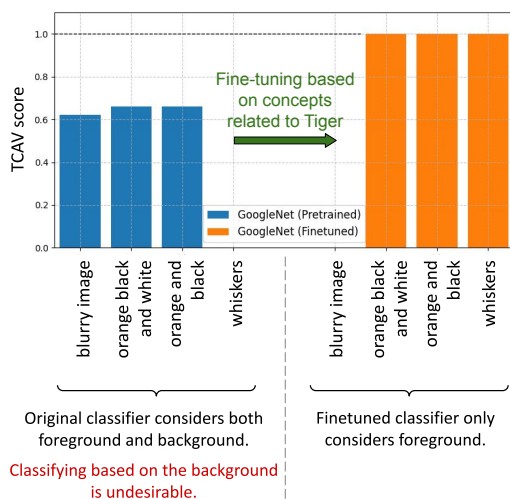

Figure 7: Usefulness of RLPO. Fine-tuning GoogleNet based on generated concepts for the Tiger class.

### 4.6. Can concept Generation Help in Understanding Model Bias?

Apart from generating actionable concepts, our proposed framework can also be used to identify spurious correlations

or undesirable biases learned by the model during training. To demonstrate this, we applied RLPO to a ResNet-18 classifer trained on the CelebA dataset (Liu et al., 2015) for "Blonde" versus "Not Blonde" classification, a task with known gender biases (De Coninck et al., 2024; Subramanyam et al., 2024; Chen et al., 2023). We adapted our seed prompts to include higher-level semantic about gender, and as shown in Fig. 8, RLPO showed that concepts representing "female face" were more important for the model for the "Blonde" class. Additionally, we see that for the "male face" seed prompt, the generated concepts started generating males with long blonde hair, further narrowing down on the fine-grain concepts the model is looking at (i.e. long and blonde hair).

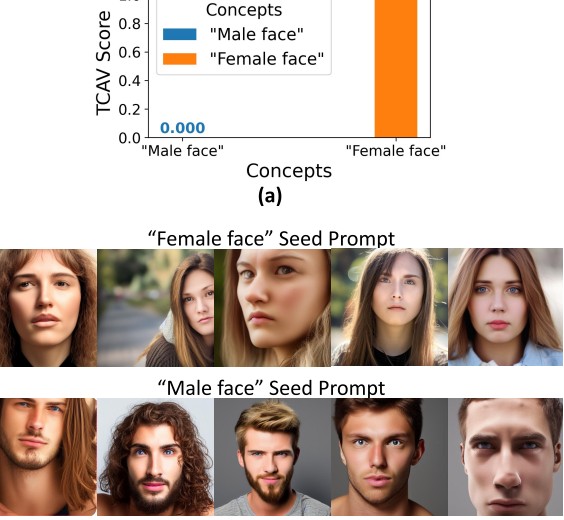

Figure 8: (a) TCAV scores obtained from "Male face" and "Female face" seed prompt, highlighting the importance of female gender over male. (b) Sample generated concepts for "Female face" and "Male face" seed prompt.

### 4.7. Can RLPO Generalized Beyond Images?

To demonstrate the generalizability of the proposed algorithm, we extended RLPO to generate words to explain sentiment analysis in NLP. We made use of Mistral-7B Instruct model to generate synonyms of seed prompts and optimized the language model based on preferences from TextCNN model pre-trained on IMDB sentiment dataset. Fig. 9 highlights relevant words with their importance score in the input. More details are provided in Appendix C.7.

## 5. Limitations and Conclusions

In this work, we introduced Reinforcement Learning-based Preference Optimization (RLPO) to automatically articulate concepts that explain the internal representations of neural networks. We demonstrated how RLPO can guide a vision-

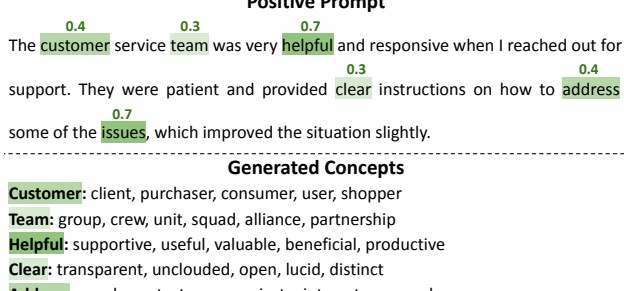

Figure 9: We use sentiment analysis in NLP to show the generalizability of RLPO. Concepts tend to be synonyms and the numbers indicate the TCAV scores.

language generative model to navigate an infinitely large concept space, addressing the challenge of manually curating concept image sets. However, our approach also suffers from some limitations. Firstly, while RLPO is designed to iteratively refine these prompts and can drift towards more model-relevant abstractions beyond the original scope, the quality and nature of the initial seeds can influence the search trajectory and efficiency. Increasing the number of seeds helps generate better concepts. Additionally, the quality and diversity of the generated concepts are also influenced by the generative model being used. The pre-training data of such models can introduce biases, and their capacity to accurately render certain concepts. While this dependency is not unique to generative approach, retrieval-based methods are similarly constrained by available data and human cognitive biases, explicitly addressing and potentially mitigating these inherited biases in the context of XAI is an important direction.

The concepts that our algorithm generates can be diverse as it tries to reveal the concepts inherent to the $f(\cdot)$, making it less domain-specific (e.g., for a medical application, there is a chance it might generate non-medical images if the $f(\cdot)$ activations get excited for non-medical data). As a future extension, we aim to input preferences from both TCAV and domain experts while optimizing, making generated explanations even more aligned to specific applications. Despite these challenges, our results demonstrate how to leverage the strengths of visual representations and adaptive learning to provide intuitive and effective solutions for understanding complex, high-level concepts in neural networks.

## Impact Statement

Our paper is a generic algorithmic contribution, and we do not foresee direct negative societal impacts. As a positive trait, our approach can potentially be used to understand and explain model bias.

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

# Appendix

In this appendix, we describe the theorems, algorithm, metrics, experimental setup and additional results.

## A. Definition, Theorems, and Proves

**Comparative Overlap of Human-Interpretable and Generative Model Concepts** in Neural Understanding Tasks ($f(\cdot)$). We formalize this with reference to Fig. 1.

**Theorem 1.** *Let the set of human-interpretable concepts that the $f(\cdot)$ has learned be $\mathcal{C}_N$, and the concept sets human collected, retrieved though segmentation, and generated using a generative model be $\mathcal{C}_H$, $\mathcal{C}_R$, and $\mathcal{C}_G$, respectively. Then, $|\mathcal{C}_G \cap \mathcal{C}_N| \geq |\mathcal{C}_H \cap \mathcal{C}_N| \geq 0$ and $|\mathcal{C}_G \cap \mathcal{C}_N| \geq |\mathcal{C}_R \cap \mathcal{C}_N| \geq 0$.*

*Proof sketch.* $\mathcal{C}_H \subseteq \mathcal{C}_G$ and $\mathcal{C}_R \subseteq \mathcal{C}_G \implies |\mathcal{C}_H \cap \mathcal{C}_N| \geq 0$ and $|\mathcal{C}_R \cap \mathcal{C}_N| \geq 0$ ☐

*Proof of Theorem 1.*

**Definition 3.** *Let $\mathcal{C}_H$, $\mathcal{C}_R$, and $\mathcal{C}_G$ denote the sets representing human-interpretable concepts, retrieved concepts, and concepts generated by a generative model, respectively. We define the relationships between these sets as follows:*

$$\mathcal{C}_H \subseteq \mathcal{C}_G \quad and \quad \mathcal{C}_R \subseteq \mathcal{C}_G.$$

**Property 1.** *For any set $\mathcal{C}_i$, where $i \in \{H, R, G\}$, it holds that:*

$$\emptyset \subseteq (\mathcal{C}_i \cap \mathcal{C}_N) \subseteq (\mathcal{C}_i \cup \mathcal{C}_N),$$

*where $\mathcal{C}_N$ represents the set of concepts learned by $f(\cdot)$.*

For any two sets $\mathcal{A}$ and $\mathcal{B}$, the size of their intersection $|\mathcal{A} \cap \mathcal{B}|$ is non-negative since it represents the number of elements common to both sets. Thus, we have:

$$|\mathcal{C}_H \cap \mathcal{C}_N| \geq 0 \quad and \quad |\mathcal{C}_R \cap \mathcal{C}_N| \geq 0. \tag{4}$$

Given Definition 3 and Property 1, we assume the following subset relationships between the sets:

$$|\mathcal{C}_H \subseteq \mathcal{C}_G| \quad and \quad |\mathcal{C}_R \subseteq \mathcal{C}_G|.$$

Case 1. Since $\mathcal{C}_H \subseteq \mathcal{C}_G$, any element $x \in \mathcal{C}_H$ is also in $\mathcal{C}_G$. Therefore, any element $x \in \mathcal{C}_H \cap \mathcal{C}_N$ is also in $\mathcal{C}_G \cap \mathcal{C}_N$. Hence,

$$\mathcal{C}_H \cap \mathcal{C}_N \subseteq \mathcal{C}_G \cap \mathcal{C}_N.$$

Case 2. Similarly, since $\mathcal{C}_R \subseteq \mathcal{C}_G$, any element $x \in \mathcal{C}_R$ is also in $\mathcal{C}_G$. Therefore, any element $x \in \mathcal{C}_R \cap \mathcal{C}_N$ is also in $\mathcal{C}_G \cap \mathcal{C}_N$. Hence,

$$\mathcal{C}_R \cap \mathcal{C}_N \subseteq \mathcal{C}_G \cap \mathcal{C}_N.$$

From Case 1 and Case 2, since $\mathcal{C}_H \cap \mathcal{C}_N$ and $\mathcal{C}_R \cap \mathcal{C}_N$ are subsets of $\mathcal{C}_G \cap \mathcal{C}_N$, it follows that:

$$|\mathcal{C}_H \cap \mathcal{C}_N| \leq |\mathcal{C}_G \cap \mathcal{C}_N| \tag{5}$$

and

$$|\mathcal{C}_R \cap \mathcal{C}_N| \leq |\mathcal{C}_G \cap \mathcal{C}_N| \tag{6}$$

Combining Eqs. 5 and 6 with the non-negativity established in Eq 4, we have:

$$|\mathcal{C}_G \cap \mathcal{C}_N| \geq |\mathcal{C}_H \cap \mathcal{C}_N| \geq 0 \tag{7}$$

and

$$|\mathcal{C}_G \cap \mathcal{C}_N| \geq |\mathcal{C}_R \cap \mathcal{C}_N| \geq 0. \tag{8}$$

☐

**The Effects of DQN-PO based Concept Space Traversal**. We now formalize what concepts the DQN has learned, with reference to Fig. 2.

**Theorem 2.** *When traversing in the concept space, with each reinforcement learning step,*

1. *Case 1: Moving from a proxy state towards an explainable state monotonically increases the reward.*
2. *Case 2: Moving from an explainable state towards the target class does not increase the reward.*

*Proof sketch.* Obtain the rewards before and after $\eta$ and compute the difference in reward for each segment. □

**Property 2.** *As $\xi$ increases, the reward function proportionally amplifies, particularly enhancing the significance of outcomes near $t_\eta$, which marks the point beyond which TCAV scores are always 1.*

*Proof of Theorem 2.* WLOG, let the TCAV score, $S_{c,m,t}$, for concept $c$ and class $m$ at time $t$ be $S_t$. The reward function is defined as (Section. 3.3),

$$R(t, a) = K \cdot S_t \cdot f(t), \tag{9}$$

for a constant $K$ and a factor,

$$f(t) = \begin{cases} \xi \cdot t & \text{if } t \le t_\eta, \\ \xi_0 & \text{otherwise,} \end{cases} \tag{10}$$

for positive parameters $\xi$ and $\xi_0$.

Case 1: Considering the difference in reward function at time $t$ when $t \le t_\eta$,

$$\begin{aligned} R(t+1, a) - R(t, a) &= K \cdot S_{t+1} \cdot f(t+1) - K \cdot S_t \cdot f(t) \tag{11} \\ &= K \cdot S_{t+1} \cdot \xi \cdot (t+1) - K \cdot S_t \cdot \xi \cdot t \\ &= K \cdot \xi \cdot (S_{t+1} \cdot (t+1) - S_t \cdot t) \\ &= K \cdot \xi \cdot (t(S_{t+1} - S_t) + S_{t+1}) \end{aligned}$$

$$\text{From } (S_{t+1} - S_t) = \frac{h(t+1) - h(t)}{(t+1) - t}) = h'(t), \tag{12}$$

$$R(t+1, a) - R(t, a) = K \cdot \xi \cdot (t \cdot h'(t) + S_{t+1}). \tag{13}$$

Since,

1. $S_t$ is monotonically increasing for $t \le t_\eta \implies h'(t) > 0$ and
2. $S_t \in [0, 1]$,

$$R(t+1, a) - R(t, a) \ge 0. \tag{14}$$

Case 2: Considering the same difference in rewards for $t \ge t_\eta$.

$$\begin{aligned} R(t+1, a) - R(t, a) &= K \cdot S_{t+1} \cdot f(t+1) - K \cdot S_t \cdot f(t) \tag{15} \\ &= K \cdot S_{t+1} \cdot \xi - K \cdot S_t \cdot \xi_0 \\ &= K \cdot \xi_0 \cdot (S_{t+1} - S_t) \end{aligned}$$

Given that $S_t$ and $S_{t+1}$ are both outcomes generated from a generative model fine-tuned for a particular concept, $S_{t+1} - S_t \approx 0$ in response to the same action. Hence,

$$R(t+1, a) - R(t, a) \approx 0. \tag{16}$$

□

Theorem 2 characterizes the how TCAV scores (i.e., proportional to rewards) are increased up to $\eta$. As a result, as shown in Theorem 3, if the generator moves close to the image class, then the explainer generates images similar to the class. Therefore, by varying $\eta$ we can generate concepts with different levels of abstractions.

**Theorem 3.** *As we go closer to the concept class, $|\mathcal{C}_G \cap \mathcal{C}_N|$ becomes larger for generated concepts $\mathcal{C}_G$ and $f(\cdot)$'s internal concepts, $\mathcal{C}_N$.*

*Proof sketch.* Measure the sensitivity difference between $S_{c_1,m,t}$ and $S_{c_2,m,t}$ as $t \to \infty$. $\qquad\square$

*Proof of Theorem 3.* At each time step $t$, two sets of samples are generated near $\mathcal{C}_G(t)$ using a generative function $g(.)$, denoted by $s_1(t) = g(\mathcal{C}_G(t))$ and $s_2(t) = g(\mathcal{C}_G(t))$. We define the sensitivity of these samples to the concept class using a measurable attribute, $\sigma(s)$, that quantifies the alignment or closeness of a sample $s$ to the target concept class.

The optimization step at each time step selects the sample with higher sensitivity, denoted by:

$$s_{\text{opt}}(t) = \arg\max\{\sigma(s_1(t)), \sigma(s_2(t))\}$$

The sample with the lower sensitivity is given by

$$s_{\min}(t) = \arg\min\{\sigma(s_1(t)), \sigma(s_2(t))\}$$

Sample $s_{\text{opt}}(t)$ and $s_{\min}(t)$ is then used to adjust $\mathcal{C}_G$, increasing its overall sensitivity to the concept class. Consequently, the sequence of $\mathcal{C}_G$ over time evolves as:

$$\mathcal{C}_G(t+1) = (\mathcal{C}_G(t) \cup s_{\text{opt}}(t)) \setminus s_{\min}(t)$$

This process incrementally increases the sensitivity of $\mathcal{C}_G(t+1)$ to the concept class, driven by the iterative inclusion of optimized samples.

Given that $\mathcal{C}_N$ is already close to the target concept class, the movement of $\mathcal{C}_G$ through this optimization process indirectly steers $\mathcal{C}_G$ towards $\mathcal{C}_N$. As $\mathcal{C}_G$ evolves in this manner, the overlap between $\mathcal{C}_G$ and $\mathcal{C}_N$ naturally increases, leading to:

$$\lim_{t \to \infty} |\mathcal{C}_G(t) \cap \mathcal{C}_N| \implies \lim_{t \to \infty} \mathcal{C}_G(t) = \mathcal{C}_N.$$

This results from $\mathcal{C}_G(t)$ containing more elements that exhibit higher sensitivity similar to those in $\mathcal{C}_N$, thereby increasing their intersection. $\qquad\square$

### A.1. Definitions

**Entropy**: Entropy quantifies the uncertainty or randomness inherent in a probability distribution. For a discrete random variable $X$ with possible outcomes $x_1, x_2, \ldots, x_n$ and corresponding probabilities $P(X = x_i) = p_i$, the entropy $H(X)$ is defined as: $H(X) = -\sum_{i=1}^{n} p_i \log p_i$, where $p_i$ represents the probability of outcome $x_i$.

**Odds**: Odds describe how many times an event is expected to happen compared to how many times it is not. They are often used in gambling, sports betting, and statistics. The odds of an event with probability p (where p is the probability of the event happening) are calculated as: $\frac{p}{1-p}$.

**Exploration Gap (EG)**: quantifies the proportion of missed optimal actions, defined as $1 - $ Accuracy, highlighting how humans frequently miss the most optimal actions when presented with generated concepts.

**Average Normalized Count (ANC)**: The ANC is a measure of the central tendency of the normalized action frequencies within a distribution. It provides insight into how the actions are distributed relative to the overall frequency distribution. A high ANC indicates that, on average, the action frequencies are relatively large, meaning that certain actions are more dominant. Conversely, a low ANC suggests that the actions are low and only a few high frequent actions are present. Given by $\frac{1}{n \cdot \max(f)} \sum_{i=1}^{n} f_i$, where $f_i$ is the frequency of action $i$.

**Inverse Coefficient of Variation (ICV)**: A standardized measure of concentration, calculated as the ratio of the mean to the standard deviation: $\frac{\mu}{\sigma}$. It represents how many standard deviations fit into the mean.

**Feedback Cost**: Feedback Cost refers to the resource(GPU) expense associated with obtaining feedback during the training of the model.

**Execution Time**: Execution time refers to the total time taken by a model or algorithm to complete its task from start to finish. This includes the time for data processing, model computation, and generating outputs.

## B. Methodology

### B.1. Machine Learning Models We Use

**Neural Network Under Test** ($f(\cdot)$): We test RLPO for all the different classification models given below.

1. GoogleNet: We utilized a pretrained model from PyTorch torchvision pretrained models with weights initialized from GoogLeNet_Weights.IMAGENET1K_V1.
2. InceptionV3: We utilized a pretrained model from PyTorch torchvision pretrained models with weights initialized from Inception_V3_Weights.IMAGENET1K_V1.
3. Vision Transformer (ViT): We utilized a pretrained model from PyTorch torchvision pretrained models with weights initialized from ViT_B_16_Weights.IMAGENET1K_V1.
4. Swin Transformer: We utilized a pretrained model from PyTorch torchvision pretrained models with weights initialized from Swin_V2_B_Weights.IMAGENET1K_V1.
5. TextCNN sentiment classification model: We utilized a pretrained model from Captum library. The model was trained on IMBD sentiment dataset.

**TCAV Logistic Model**: We utilized a logistic regression model to address classification tasks in TCAV instead of the default SGD (Stochastic Gradient Descent) classifier. This decision was based on our observation that the SGD classifier produced high variance TCAV (Testing with Concept Activation Vectors) scores, which indicated inconsistent model behavior across different runs. As demonstrated in Table 7, the standard deviations for SGD-derived scores are frequently larger than those obtained with the logistic model, underscoring the latter's improved stability. We configured the model to perform a maximum of 1000 iterations (max_iter=1000).

Table 7: TCAV Scores (Concept/Random) for Different Models, Layers, and Classifiers. Scores are presented as mean $\pm$ standard deviation across 5 runs with random seed. Concepts used in these experiments are from the ones provided by the authors of TCAV paper.

| Model | Layer | Stripes/Random TCAV Score | | Dots/Random TCAV Score | |
| --- | --- | --- | --- | --- | --- |
| | | SGD | Logistic | SGD | Logistic |
| GoogleNet | inception3a | $0.662 \pm 0.03 / 0.338 \pm 0.03$ | $0.67 \pm 0.00 / 0.33 \pm 0.00$ | $0.36 \pm 0.05 / 0.64 \pm 0.05$ | $0.33 \pm 0.00 / 0.67 \pm 0.00$ |
| | inception4e | $0.992 \pm 0.01 / 0.008 \pm 0.01$ | $1.00 \pm 0.00 / 0.00 \pm 0.00$ | $0.01 \pm 0.007 / 0.99 \pm 0.007$ | $0.00 \pm 0.00 / 1.00 \pm 0.00$ |
| ResNet50 | layer3 | $0.796 \pm 0.02 / 0.204 \pm 0.02$ | $0.78 \pm 0.00 / 0.22 \pm 0.00$ | $0.078 \pm 0.07 / 0.922 \pm 0.07$ | $0.00 \pm 0.00 / 1.00 \pm 0.00$ |
| | layer4 | $1.000 \pm 0.00 / 0.000 \pm 0.00$ | $1.00 \pm 0.00 / 0.00 \pm 0.00$ | $0.60 \pm 0.54 / 0.40 \pm 0.54$ | $0.00 \pm 0.00 / 1.00 \pm 0.00$ |

**Stable Diffusion v1-5 with LoRA**: We used our base generation model as SD v1-5 and updated its weights using LoRA during preference optimization step. This version of SD was finetuned from SD v1-2 on "laion-aesthetics v2 5+" dataset with 10% drop in text-conditioning for better CFG sampling. In our experiments, we kept LoRA rank to 8 with a scaling factor of 8 and initial weights were defined from a gaussian distribution. We only targeted the transformer modules of U-Net in the SD architecture.

**DQN**: We use a DQN with specific parameters tailored to effectively navigate a vast search space. We utilized a small buffer size of 100, which limits the number of past experiences the model can learn from, encouraging more frequent updates. The exploration rate was set at 0.95, prioritizing exploration significantly to ensure thorough coverage of the search space. The batch size was configured to 32. We set the discount factor to 0.99 and the update frequency was set at every four steps. The model updates its parameters with a the soft update coefficient of 1.0. Gradient steps was set to 1 indicating a single learning update from each batch, and gradient clipping was capped at 10 to prevent overly large updates.

**BLIP**: We utilize the Bootstrapped Language Image Pretraining (BLIP) model for the task of Visual Question Answering (VQA). This model, sourced from the pre-trained version available at 'Salesforce/blip-vqa-capfilt-large', is designed to generate context-aware responses to visual input by leveraging both image and language understanding. The large variant of

the BLIP model is fine-tuned for VQA, allowing it to effectively interpret and answer questions based on the visual content provided.

## B.2. RLPO Algorithm

Algorithm 2 presents the detailed version of the algorithm introduced in Section 3, Algorithm 1.

---

**Algorithm 2** DQN Algorithm with DPO and Adaptive Reward

---

**Input:** Set of test images, $f(\cdot)$
Initialize Q-network $Q_\theta(s, a)$ with random weights $\theta$
Initialize replay buffer $\mathcal{D}$ and adaptive parameter $\xi \leftarrow 0.1$
**for** each episode **do**
    **for** each time step $t$ **do**
        Observe state $s_t$ and select action $a_t$ based on $Q$ ($\epsilon$-greedy)
        Execute $a_t$ and generate 10 images, divided into two groups $G_1$ and $G_2$
        Evaluate TCAV scores $\overline{TCAV}_1$ and $\overline{TCAV}_2$
        **if** $\max(\overline{TCAV}_1, \overline{TCAV}_2) \leq 0.7$ **then**
            Update policy to favor higher $\overline{TCAV}$ group and perform DPO
            Update $\xi \leftarrow \min(1, \xi + \text{increment})$
        **else**
            Set $\xi \leftarrow 1$
        **end if**
        Compute reward $r_t = \xi \cdot \max(\overline{TCAV}_1, \overline{TCAV}_2)$
        Store transition $(s_t, a_t, r_t, s_{t+1})$ in $\mathcal{D}$
        Sample a mini-batch from $\mathcal{D}$
        **for** each sampled transition $(s_i, a_i, r_i, s_{i+1})$ **do**
            Compute target $y_i = r_i + \gamma \max_{a'} Q_{\theta'}(s_{i+1}, a')$
        **end for**
        Compute loss $L(\theta) = \frac{1}{N} \sum_{i=1}^{N} (y_i - Q_\theta(s_i, a_i))^2$
        Perform a gradient descent step to update $\theta$
        Periodically update target network: $\theta' \leftarrow \tau\theta + (1 - \tau)\theta'$
    **end for**
**end for**
**Output:** Set of concept images

---

## B.3. Preprocessing for Generating the Action Space

Steps not discussed in Section 3.2.

Each patch from the test images is passed to the VQA model to extract relevant and useful information about the corresponding class. In this study, we choose BLIP (Li et al., 2022) as our VAQ model. We posed a set of targeted questions to the VQA model, aiming to gain insights into the class-specific features represented in the patches. The questions are designed to probe various aspects of the image patches, helping the model focus on class-defining attributes.

1. "What is the pattern in the image?"
2. "What are the colors in the image?"
3. "What is the background color of the image?"
4. "What is in the background of the image?"
5. "What is the primary texture in the image?"
6. "What is the secondary texture in the image?"
7. "What is the shape of the image?"

We then remove stop words and duplicates from the generated responses using lemmantizing and perform a cross-similarity check using CLIP between all the unique words and further filtered words which are more than 95% similar. To further select most relevant keywords to the class images, we perform a VLM check using class images and the extracted keyword to get the softmax score of how much the keyword and image are related. This score is then averaged over all the class

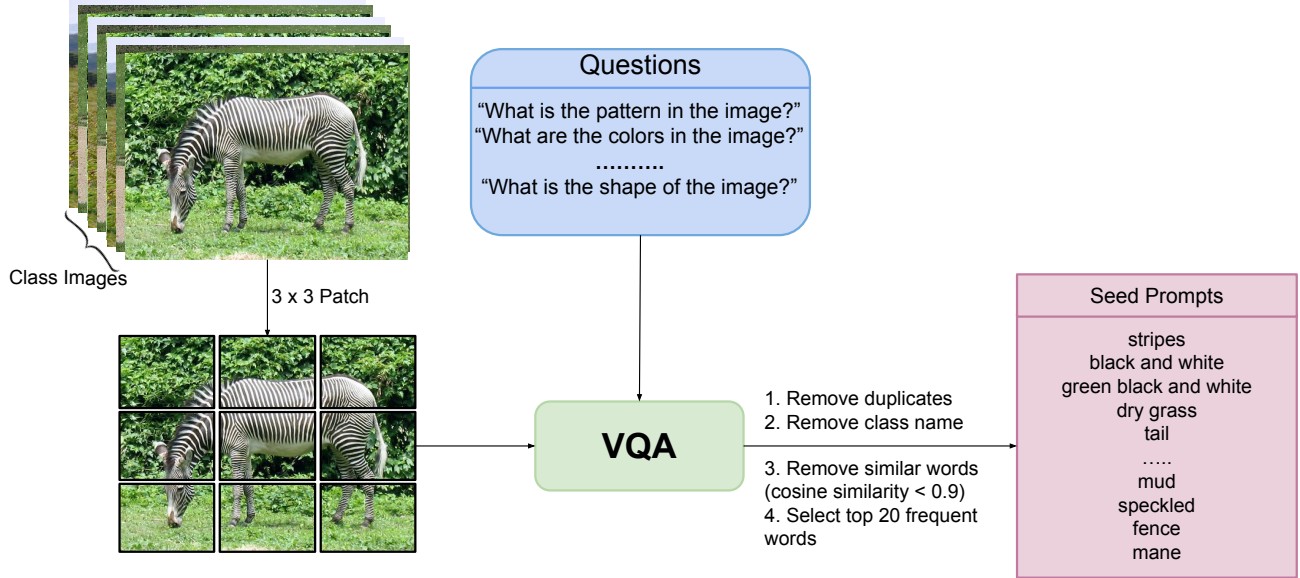

Figure 10: Seed prompt pipeline

images and this average is use to sort the keywords. Now, from the sorted keywords, we select top 20 keywords as our RL action space. The cross-similarity and VLM check are inspired from (Zang et al., 2025) where they used a similar filtering setup to remove potentially useless concepts.

### B.4. Exploration of Random Gibberish Prompts as Seed Prompts

In this experiment we don't use a VQA to get seed prompts. We choose a random list of incoherent prompts, example shown in Fig. 11. We found that for these prompts it takes a really long time to get some meaningful explanation and in most cases lead to random generation, thus showing importance of starting from good proxy concepts.

> Gibberish Seed Prompts
>
> 1. dKgN MTW8bvbxB6aW1L2TfTuTYZK3He0urbEEmclEpY
> 2. se-L8fPe19ZzUmuM uDYVYusFnYtNZeFM1YqXdE57Y7OMD3Z80cKwLo5
> 3. CzKLTlZZnWHjtBn80wIfC z8O
> 4. mhtxqyH2FBEC
> 5. SWEC6Wlqfpqaz PQjoGrxIuzm m2ua8oGJySIeG2NqCG9BBvU9Eerj7wheWk7j-t

Figure 11: Sample gibberish seed prompt used with RLPO on GoogleNet classifier to generate concepts.

### B.5. Preference Optimization Update for State Space

Steps not discussed in Section 3.4.

The candidate concepts serve as the initial states for the RL agent. From these initial states, the agent takes actions $a \in$ Keywords that leads to multiple subsequent possible states using $g(.)$. These states are then grouped, and the group's sensitivity is compared against Inputs of $f(\cdot)$ using TCAV scores. A higher TCAV score suggests higher sensitivity, indicating that the group is more aligned with $f(\cdot)$'s inputs.

We employ preference optimization over the grouped states to guide states towards explainable concepts. To prevent the model from skipping over explainable states and directly reaching the input domain, we introduce a threshold that limits the application of preference optimization at each step as shown in equation 17.

Given two groups of samples $G_1$ and $G_2$ with their average TCAV scores $\overline{TCAV}_1$ and $\overline{TCAV}_2$ :

$$\text{if } \max(\overline{TCAV}_1, \overline{TCAV}_2) \leq 0.7, \text{ update } \pi \text{ to favor the group with higher } \overline{TCAV}. \tag{17}$$

To optimize $g(.)$ to find better proxies, for each step in the environment we utilized average TCAV scores $\overline{TCAV}_1$ and $\overline{TCAV}_2$ from $G_1$ and $G_2$ to decide between preferred and unpreferred concepts. Lets say $\overline{TCAV}_1 \succ \overline{TCAV}_2$, than we optimize $g(.)$ over the sample S defined as $S = \{(a, x_0^{g1}, x_0^{g2})\}$, where $x_0^{g1}$ and $x_0^{g2}$ are the sample points from the groups on action $a$. We optimize $g(.)$ using objective 18 to get a new optimzed $g'(.)$ (Wallace et al., 2024).

$$L(\theta) = - \mathbb{E}_{(x_0^{g1}, x_0^{g2}) \sim S, t \sim U(0,T), x_t^{g1} \sim q(x_t^{g1}|x_0^{g1}), x_t^{g2} \sim q(x_t^{g2}|x_0^{g2})} \log \sigma \left( -\beta T \omega(\lambda_t) \right.$$
$$\left( \|\epsilon^{G1} - \epsilon_{g'(.)}(x_t^{G1}, t)\|_2^2 - \|\epsilon^{G1} - \epsilon_{g(.)}(x_t^{G1}, t)\|_2^2 \right.$$
$$\left. \left. - \left( \|\epsilon^{G2} - \epsilon_{g'(.)}(x_t^{G2}, t)\|_2^2 - \|\epsilon^{G2} - \epsilon_{g(.)}(x_t^{G2}, t)\|_2^2 \right) \right) \right) \tag{18}$$

where $x_t^* = \alpha_t x_0^* + \sigma_t \epsilon^*$, $\epsilon^* \sim \mathcal{N}(0, I)$ is drawn from $q(x_t^*|x_0^*)$. $\lambda_t = \alpha_t^2/\sigma_t^2$ is the signal-to-noise ratio, and $\omega(\lambda_t)$ is weighting function (constant in practice).

### B.6. TCAV Setting for Different Models

We tested different models on different layers and classes and the summary of our setting across different models is described in table 8.

Table 8: TCAV setting across different models

| Models | Layers | ImageNet Classes |
|---|---|---|
| GoogleNet | inception4e layer | Goldfish, Tiger, Zebra & Police Van |
| InceptionV3 | Mixed_7c layer | Goldfish, Tiger, Lionfish & Basketball |
| Vision Transformer (ViT) | heads layer | Goldfish, Golden Retriever, Tiger & Cab |
| Swin Transformer | head layer | Goldfish, Jay, Siberian husky & Tiger |

## C. Experiments

### C.1. Computing Resources

The experiments were conducted on a system equipped with an NVIDIA GeForce RTX 4090 GPU, 24.56 GB of memory, and running CUDA 12.2. The system also featured a 13th Gen Intel Core i9-13900KF CPU with 32 logical CPUs and 24 cores, supported by 64 GB of RAM. This setup is optimized for high-throughput computational tasks but the experiments are compatible with lower-specification systems.

### C.2. Human and LLM-Based AI Feedback Mechanisms

We test other feedback mechanism in RL by replacing the XAI-TCAV feedback with AI and Human feedback's. Herer we discuss the experiental setup and configuration for both experiements.

**AI Feedback**: GPT-4 is leveraged to evaluate the explanatory power of image sets by focusing on concepts related to a target class, a method that aligns with the growing trend of incorporating AI-driven feedback (Bai et al., 2022). This approach involves sending a structured prompt to an LLM, asking it to score how well two sets of images explain a target class using a specified concept. The process involves the following.

1. Image Encoding: The images from two sets (concept1 and concept2) are first converted into a base64 format to ensure they can be transmitted via the request as encoded strings.
2. Structured Prompt: A detailed and specific prompt is crafted for the LLM. It asks the model to assess the quality of explanation each image set provides for a particular class through the lens of a specific concept. The prompt used is

"Please evaluate each of the following sets of images for how well they explain the class {class_name} via the concept {concept_name}. For each set, provide a numerical score between 0 and 1 (to two decimal places)" The prompt clearly defines how the model should respond, asking for a numerical score between 0 and 1, where:

  (a) 0 indicates that the image set does not explain the class at all via the concept.

  (b) 1 indicates that the image set perfectly explains the class via the concept.

3. LLM-Based Scoring: Once the prompt is sent to the LLM, it evaluates the image sets and provides scores based on its learned knowledge and understanding. The response is parsed to extract the scores for each set of images.

**Human Feedback**: In this experiment, eight computer science majors provided live feedback after each step of a reinforcement learning process, leveraging their prior knowledge of reinforcement learning with human feedback (RLHF) mechanisms (Christiano et al., 2017). The feedback from all participants was averaged to serve as the reward for each step in the RL process. Given the abstract nature of the initial concepts, participants needed to take time to thoughtfully assess each step, which contributed to a lengthier feedback cycle.

### C.3. Additional Results and Analysis

To validate our method for its ability to generate concepts, we tested it with different models and classes. We started it on traditional models, GoogleNet and InceptionV3, and then extended it to transformer-based models, Vision Transformer (ViT) and Swin Transformer, pre-trained on ILSRVC2012 data set (ImageNet) (Krizhevsky et al., 2017). We show additional plot in various classes shown in Fig 12,13,14,15,16.

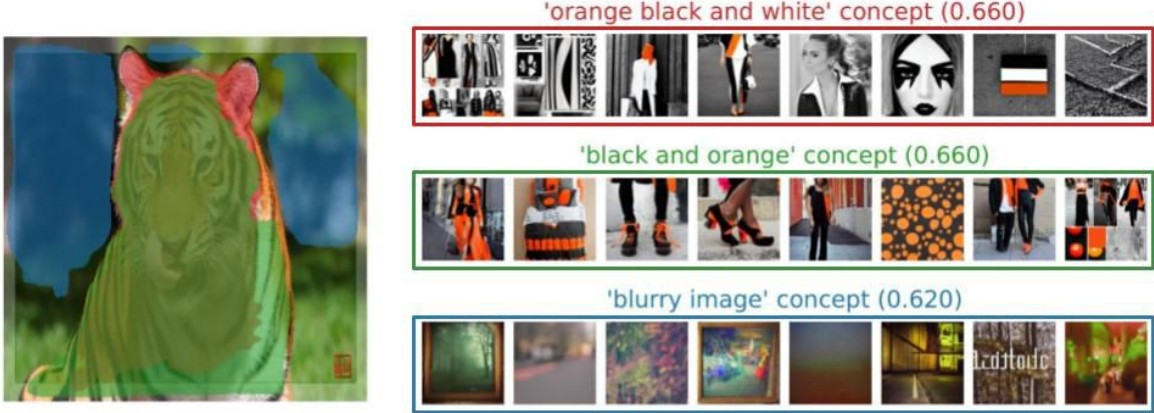

Figure 12: Explanation plot of Tiger classification by GoogleNet from RLPO.

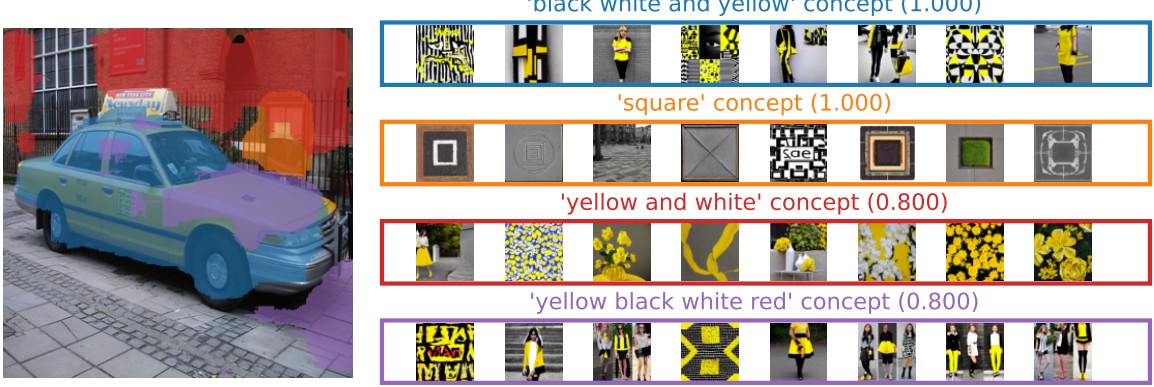

Figure 13: Explanation plot of Cab classification by ViT from RLPO.

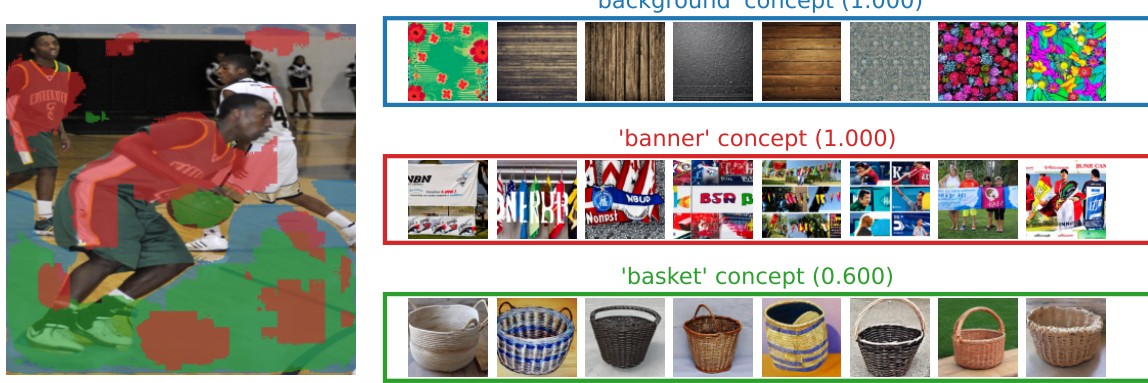

Figure 14: Explanation plot of Basketball classification by InceptionV3 from RLPO.

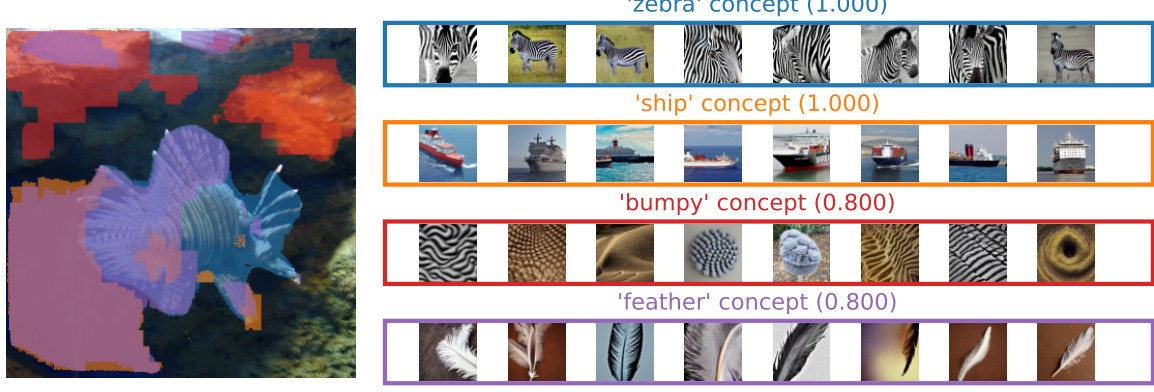

Figure 15: Explanation plot of Lionfish classification by InceptionV3 from RLPO.

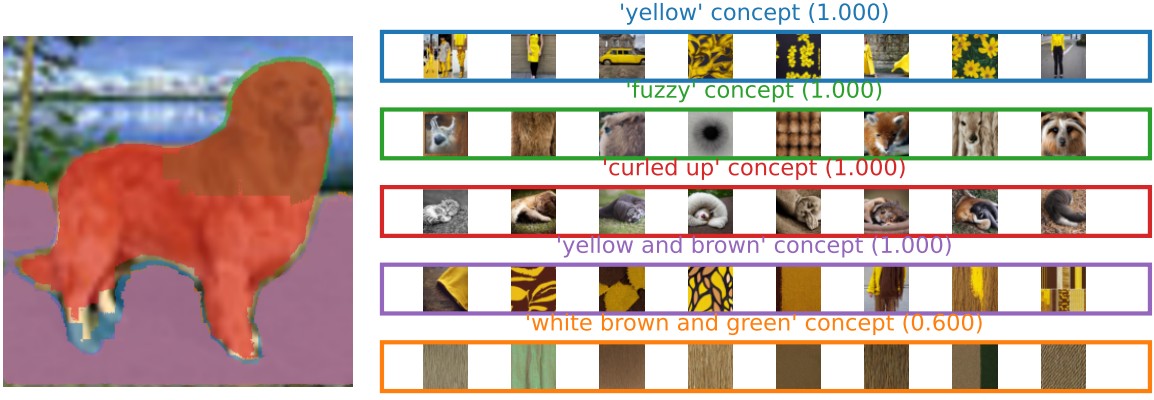

Figure 16: Explanation plot of Golden Retriever classification by ViT from RLPO.

### C.3.1. CUMULATIVE REWARDS

The cumulative rewards during training for GoogleNet and InceptionV3 is shown in Fig. 17. For ViT and Swin Transformer it is shown in Fig. 18. This figure illustrates the steady accumulation of rewards over time as they interact with the reinforcement learning environment. All models demonstrate a steady increase in cumulative rewards, the classes with higher reward peak reaches its explinable state faster.

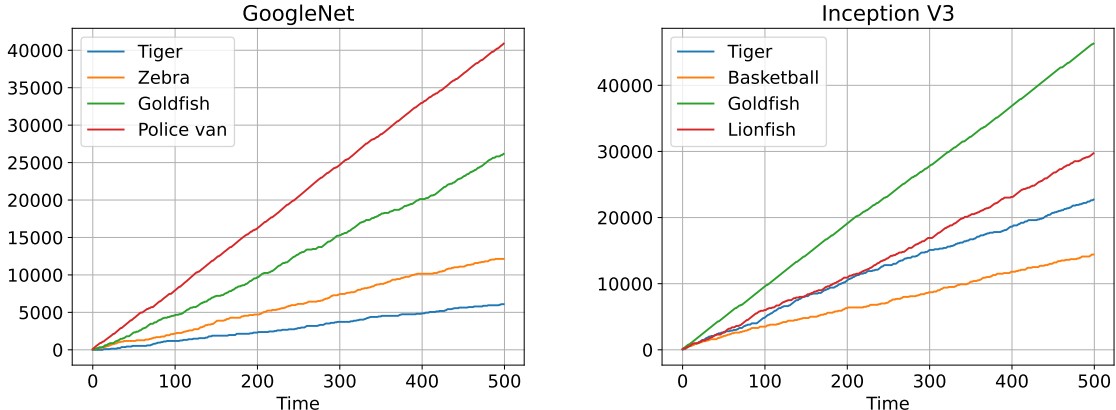

Figure 17: Cumulative rewards on traditional models.

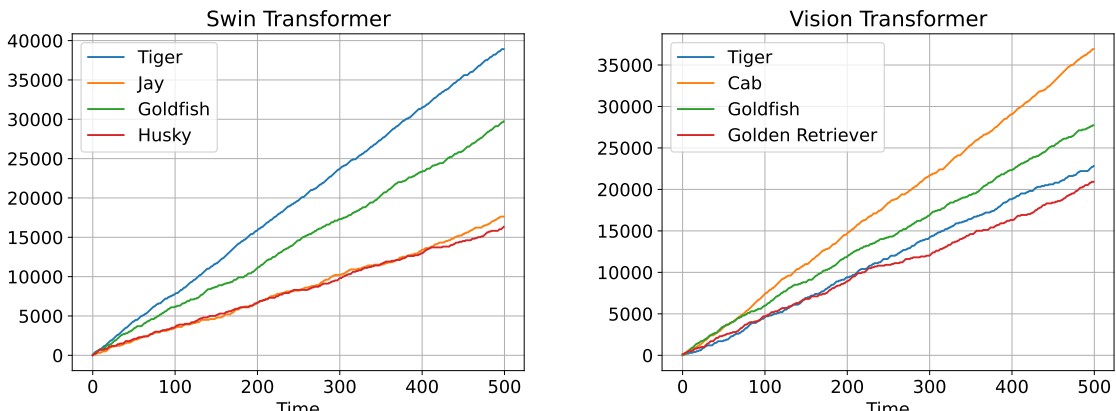

Figure 18: Cumulative rewards on transformer models.

### C.3.2. ACTION SELECTION OPTIMIZATION DURING RLPO TRAINING

As shown in Fig. 19, during training with multiple combinations of seed prompts, we observe that the RL agent initially explores various action combinations. However, as training progresses, individual actions become more optimized due to preference optimization (PO). This leads the agent to prefer fewer action combinations, since just choosing one or two actions makes the agent reach an explainable state.

### C.3.3. CONCEPT HEATMAP

To determine the relationship between generated concepts and test images, we made use of CLIPSeg transformer model (Lüddecke & Ecker, 2022). We passed generated concepts as visual prompts and test images as query images into the model and it returns a pixel-level heatmap of the probability of visual prompt in the query image. Fig. 20, 21 showcases some examples on concept heatmap indicating the presence of the concept in the image.

### C.3.4. C-DELETION

The central idea behind c-deletion in explainability is to identify and remove parts of the input context that are not crucial for the decision-making process, allowing for clearer insights into how the model arrives at its predictions or actions. C-deletion evaluations assesses the impact of removing certain contextual inputs (features, variables, or states) on a model's performance as shown in Fig. 24.

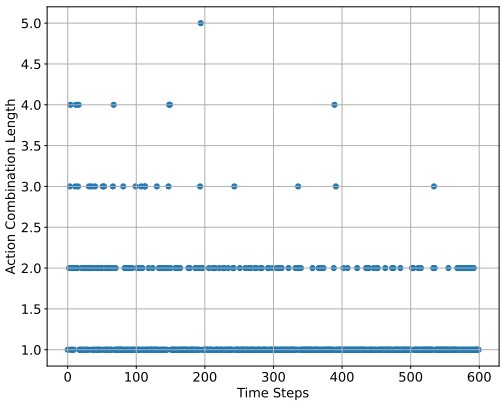

Figure 19: Combined actions (multiple keywords) count over training time

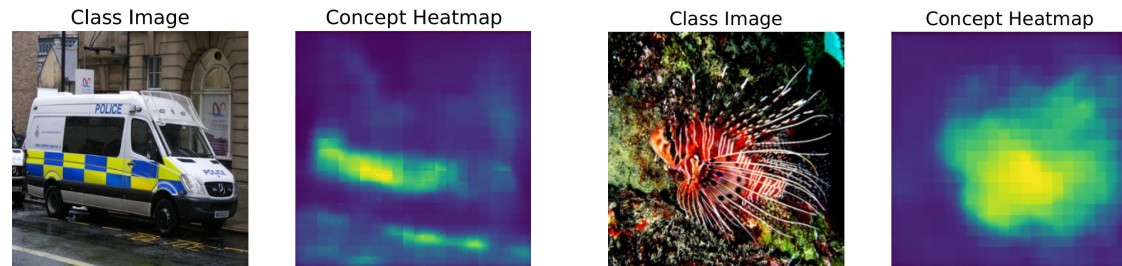

Figure 20: Van class with "white blue and yellow", Lion fish class with "zebra" seed prompt.

### C.4. Abstract Concepts

While generating concepts with RLPO, we can observe the progression of output concepts produced by the SD model. As shown in Fig. 25, applying RLPO to the seed prompt "zoo" on the tiger class reveals different levels of abstraction. These abstractions provide insight into what the model prioritizes when identifying a tiger—progressing from a four-legged orange-furred animal to one with black and white stripes, and finally to an orange-furred animal with stripes and whiskers. We obtain concepts at varying levels of abstraction by adjusting $\eta$, though our method currently cannot determine $\eta$ to achieve a specific abstraction level.

### C.5. Qualitative Comparison Between With and Without RL Preference Optimization

To demonstrate the usefulness of RL based preference optimization, we compared the output generated from the diffusion model fine-tuned using RL with the one fine-tuned iteratively over every seed prompt (brute-force). As shown in Fig. 26, if we don't use RL to optimize, we see that stripes seed does not converge to a good quality concept compared to the one obtained using RL for the same time budget. The main reason for that is, the RL agent learns over time what trajectories are worth optimizing and drops the less explainable trajectories, highlighting the need for using RL agent while optimization.

### C.6. Qualitative Comparison With Other Popular XAI Techniques

We compare the output generated by other popular XAI techniques such as LIME and GradCam with ones generated by RLPO. As shown in Fig. 27, we can see that other methods just explains where the model is looking at whereas our approach also explains what type of features is the model focuses on.

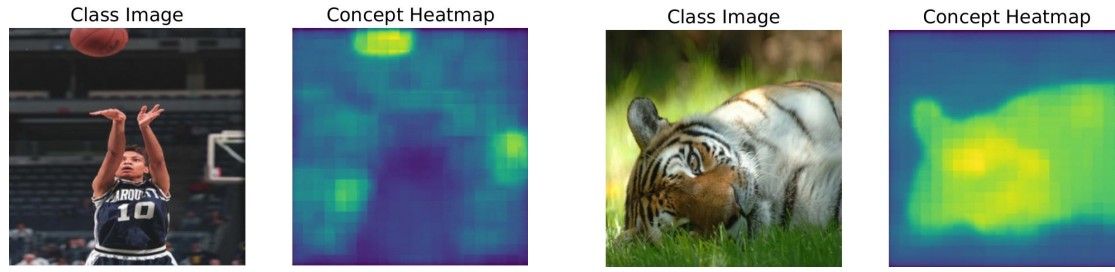

Figure 21: Basketball class with "basket" seed prompt, Tiger class with "orange black and white" seed prompt.

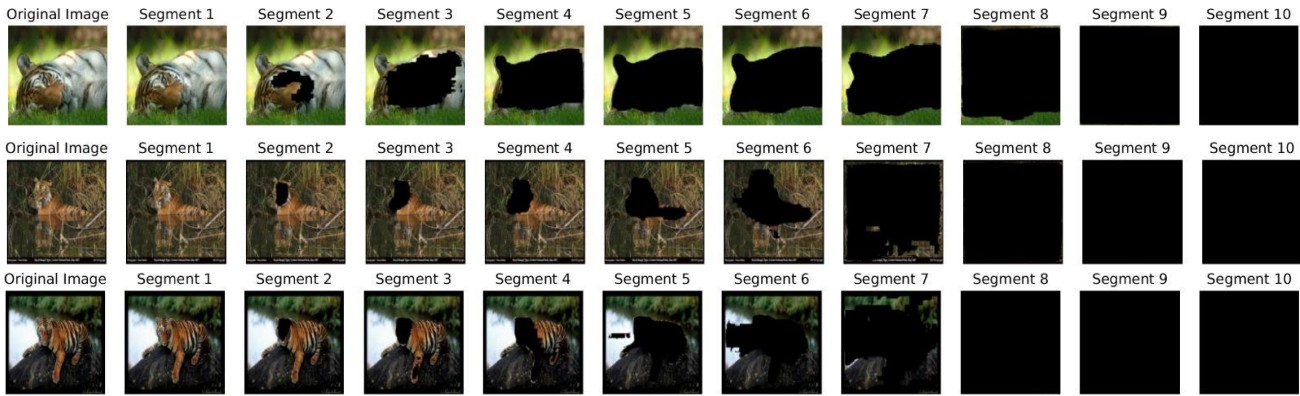

Figure 22: The figure shows c-deletion taking place for different images from "tiger" class over time for "orange black and white" seed concept.

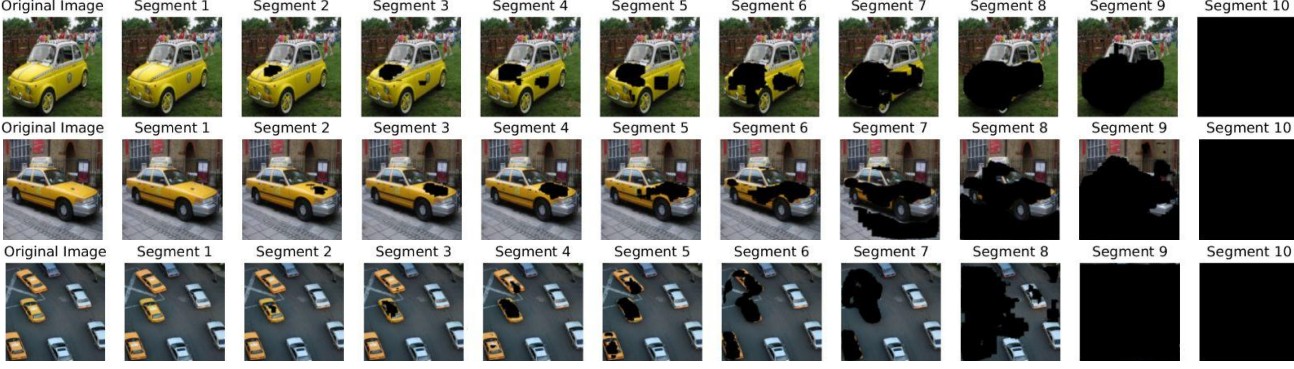

Figure 23: The figure shows c-deletion taking place for different images from "cab" class over time for "yellow and white" seed concept.

## C.7. RLPO in Sentiment Analysis

We extended our method to the NLP domain, successfully identifying which parts of the input contribute to specific outputs. For sentiment analysis, a binary classification problem, we present results for both positive and negative classes.

A list of positive and negative prompts was created, analogous to class images in traditional image classification tasks. Random prompts, similar to those in Fig. 11, were used to simulate random classes. Every word in the prompt, excluding stop words, along with its synonyms, was treated as a concept for this experiment. Synonyms were generated using the Mistral-7B Instruct model, serving a role comparable to the image generation model in image-based settings. We observed that multiple words were identified, along with their influence on the overall prompt, for both classes (positive and negative) as shown in Fig. 9 and Fig. 28.

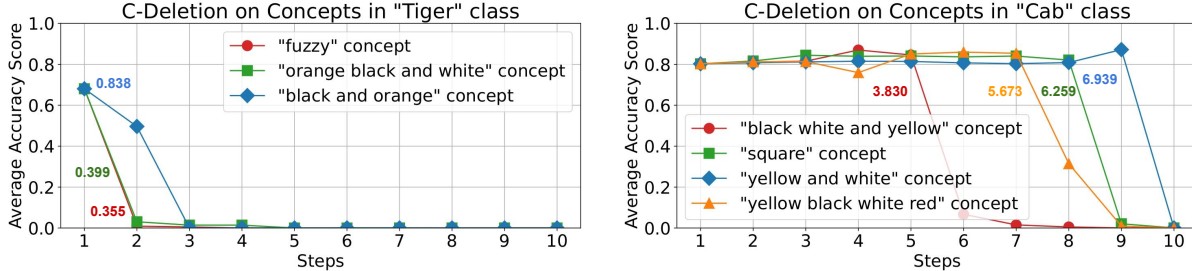

Figure 24: C-deletion. Removing concepts over time to measure the reliability. The colored numbers indicate the area under the curve (the lower the better).

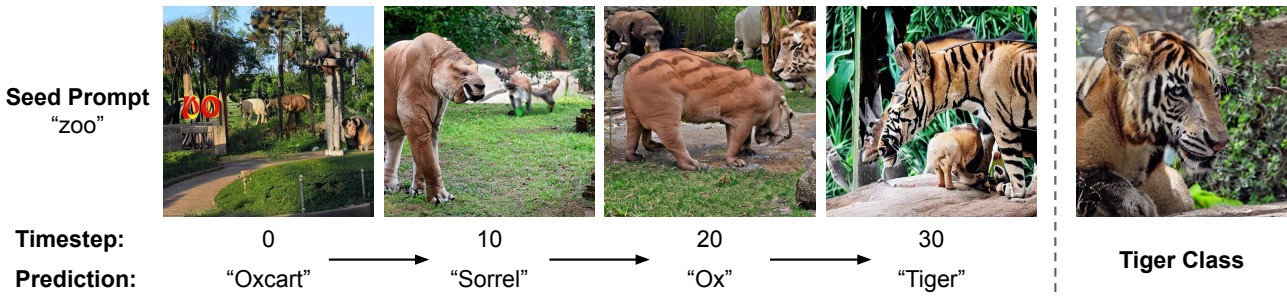

Figure 25: Different levels of abstraction for the "Tiger" class on the GoogleNet classifier are illustrated. The generated image starts as a random "zoo" image and gradually transitions to images with tiger-like features. Observe that the seed prompt "zoo" becomes more animal-like at t=10, gains more stripes at t=20, develops tiger-specific colors at t=30, and progressively refines into a tiger image. The model's prediction also evolves, starting from a random classification of "oxcart" to confidently identifying the generated concept as "tiger".

### C.8. Effects of Fine-Tuning

When fine-tuning a model, the optimization process updates its weights through gradient-based methods, causing shifts in the concepts (Fig. 7) it learns. These weight adjustments modify how the model attends to different regions or patterns in an image, leading to changes in the internal activation maps and the conceptual understanding of the input. As the model learns new concepts or refines existing ones, it adjusts its feature extraction and decision-making processes to better align with the specific objectives of the fine-tuning task, thereby altering the way it interprets and generates outputs.

To demonstrate this we conducted an experiment. In this experiment, we choose a pretrained GoogleNet classifier for the Tiger class whose important seed prompts were 'orange black and white', 'orange and black', and 'blurry image' with TCAV scores of 0.66, 0.66, and 0.62, respectively. Out of these seed prompts, 'orange black and white' and 'orange and black' highlight the tiger pixels while 'blurry image' seed prompt highlights the background pixels (see Fig. 12). This tells us that in order to classify a tiger, GoogleNet looks at both the foreground and background. Now, the engineers want the classifier to classify the tiger based on tiger pixels, not its background (note: from the classical Wolfe-Husky example in LIME (Ribeiro et al., 2016), we know the spurious correlation of background).

To this end, we generated 100 tiger images based on concepts related to 'orange black and white' and 'orange and black' using a separate generative model and fine-tuned our Googlenet model. Running RLPO on this fine-tuned model revealed that the model learned some new concepts such as 'whiskers' and also revealed that previous concepts such as 'orange black and white' and 'orange and black' are now more important with TCAV scores of 1.0 and 1.0, respectively. This means that the classifier is now only looking at tiger pixels, not the background. Dataset samples are shown in Fig. 29.

**"Stripes" concept without RL**

**"Stripes" concept with RL**

Figure 26: "Stripes" concept generation from SD models optimized with and without RL

## C.9. Human Survey: Understanding Human Capabilities

The survey involved 50 participants, each of whom was shown 10 class images along with two concept options as shown in Fig. 30: one derived from a retrieval-based method and the other generated using RLPO. The participants were divided into Laymen and Experts.

1. Expert: Computer science graduates who are familiar with the concept of explainability and have a working knowledge of AI or machine learning systems.
2. Laymen: Individuals without expertise in computer science, AI, or explainability, representing the general public's perspective.

## C.10. Human Study: Usability

We conducted a human study to evaluate the usefulness of the provided explanations, involving 19 Machine Learning (ML) engineers. Fig. 31 shows the survey used while conducting the human study. This study was approved by the Institutional Review Board of Arizona State University (IRB ID STUDY00021561).

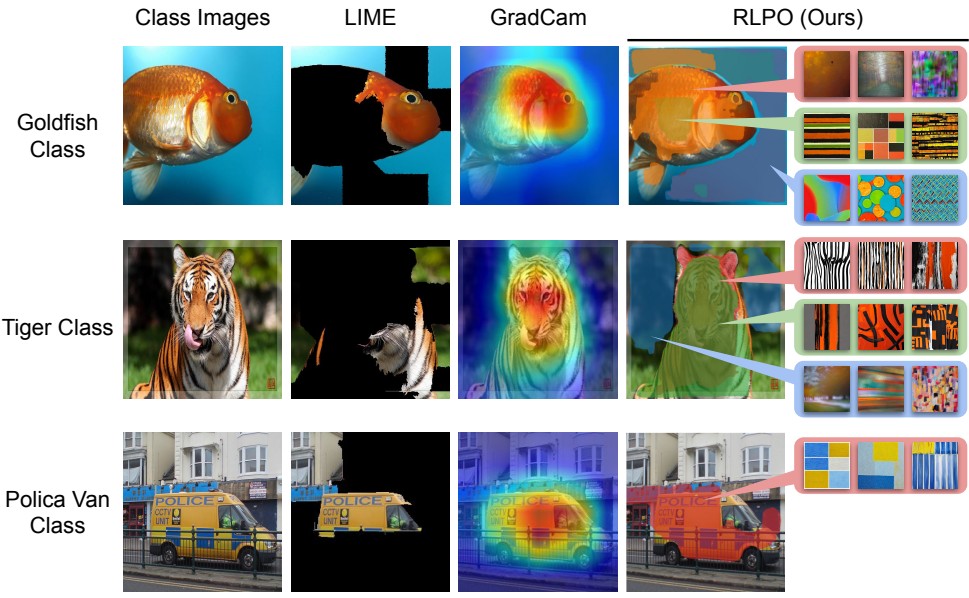

Figure 27: Comparison of concepts identified by different methods. RLPO can show the correspondences between test image and different concepts.

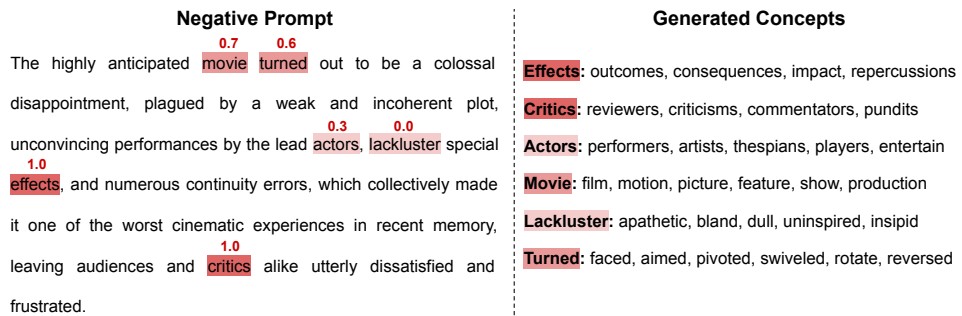

Figure 28: Generated concepts explain why a given text is classified as a negetive sentiment.

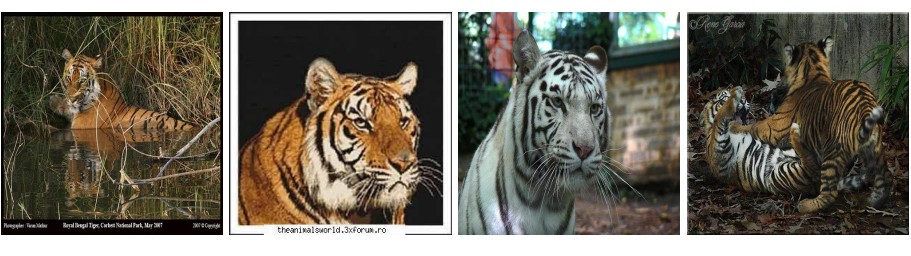

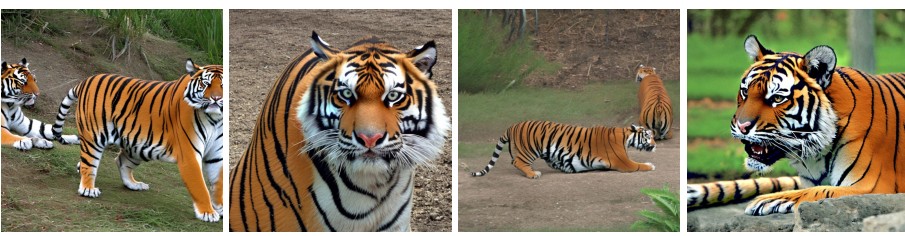

Figure 29: Dataset samples used to fine-tune GoogleNet classifier on Tiger class.

# Understanding Human Capabilities

Thank you for considering participation in our survey. Please read the following information carefully before proceeding.

When neural networks are trained using images, they identify and learn specific high-level concepts to recognize those images. Typically, as humans, we do not know what those high-level concepts are. In this survey, your task is to **guess**, among the two given options, which high-level concepts (also, represented as images) could the neural network has learned to identify each test image.

**Note: There is no one correct answer, the selection(s) are based on your belief and understanding. You can select none, one, or both concept images.**

- **Purpose of the Survey:** This survey is conducted solely for educational purposes to understand human opinions.
- **Data Use:** The data collected through this survey will not be used for training any models, algorithms, or other computational tools. The primary use of the data will be used to understand human opinion and confined to educational contexts.
- **Confidentiality:** Your responses will be treated with the utmost confidentiality. No individual data will be disclosed publicly or used outside the scope of the educational objectives stated.

Sign in to Google to save your progress. Learn more

1. Which of the following option(s) could be the reason for a neural network to classify the following image as **zebra**?

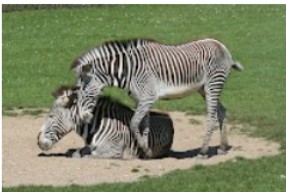

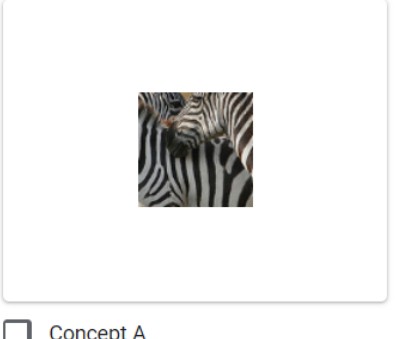

☐ Concept A

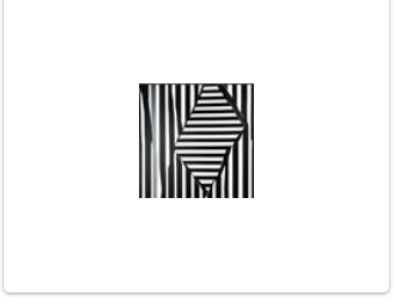

☐ Concept B

Figure 30: A screenshot from our human survey with instructions and a sample question.

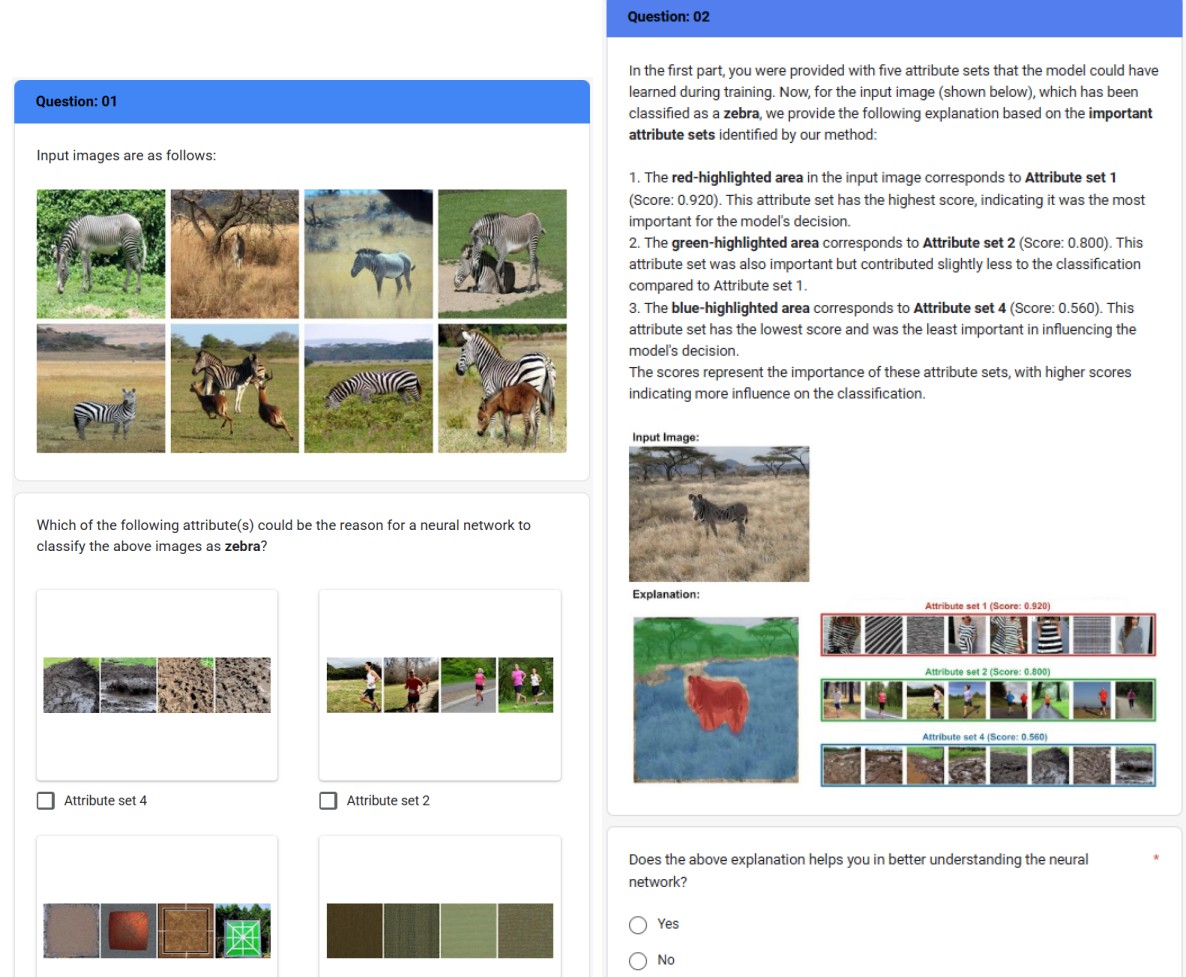

Figure 31: Screenshots from our human survey with sample questions to validate usability.

