# OpenReview forum: "Explainable Concept Generation through Vision-Language Preference Learning for Understanding Neural Networks' Internal Representations"
_ICML.cc/2025/Conference — ICML 2025 poster_

### Official Review · Reviewer_Rqpq · 2025-03-08

**Overall Recommendation:** 3

**Summary:**

The paper addresses a critical challenge in concept-based explanation methodology—specifically, the generation of "concepts" for explanations. Traditionally, this required practitioners to manually guess and collect various candidate concept image sets. The paper introduces a novel approach, utilizing reinforcement learning-based preference optimization (RLPO), to guide the Stable Diffusion model in generating concepts that are significant to the neural network's internal representations. Both qualitative and quantitative results are presented to demonstrate the effectiveness of RLPO in uncovering these representations.

**Claims And Evidence:**

In my view, the claims made in the paper are generally well-supported. However, I have reservations about the qualitative results of the method, as detailed below.

**Essential References Not Discussed:**

The relation to broader scientific literature was discussed in the Section 1 and 2 of the paper.

**Experimental Designs Or Analyses:**

Please see above for my concerns regarding the user study and the limited scope of the qualitative results.

**Methods And Evaluation Criteria:**

The RLPO's approach to encouraging diffusion models to generate truly meaningful concepts is technically sound and innovative. The authors tried to address the critical limitations of previous concept-based explanation methods.

However, I have concerns about the seed prompt acquisition. While Appendix C.3 details the prompts used to probe image patches, these seem primarily focused on low-level visual concepts. The method's ability to capture higher-level semantic concepts (e.g., gender, age) remains unclear. Additionally, it's uncertain whether the stable diffusion model can generate meaningful representations for such abstract concepts. I would appreciate the authors' perspective on this limitation.

Regarding evaluation, I have reservations about the generalizability of the results. The paper presents selected examples, predominantly featuring specific cases (zebras, tigers). While user studies were conducted, they also relied on these selected examples rather than randomly sampled cases. A more robust evaluation would involve testing with random examples across diverse scenarios.

Further evidence is needed to demonstrate the method's effectiveness across broader applications. Additionally, the paper would benefit from including and analyzing failure cases to better understand the method's limitations. For instance, the concepts produced by RLPO in Figure 4, while diverse, don't appear particularly meaningful.

**Other Comments Or Suggestions:**

N/A

**Other Strengths And Weaknesses:**

In general, I find the method proposed in the paper technically sensible, and I agree that issue tracking is crucial in concept-based explanations. However, I have reservations about the assumptions made in the paper. More importantly, I hold reservations about the qualitative results of the methods in broader cases.

**Questions For Authors:**

N/A

**Relation To Broader Scientific Literature:**

The relation to broader scientific literature was discussed in the Section 1 and 2 of the paper.

**Theoretical Claims:**

Regarding Figure 1 and its associated theorem (including proofs), the underlying assumptions should be more explicitly stated. In particular, I would to like see clearer justification for the assumption that "$C_H\subseteq C_G$ and $C_R\subseteq C_G$".

---

> ### Author Rebuttal · Authors · 2025-04-01
>
> Thank you for your thoughtful review and for recognizing the technical soundness and novelty of our framework. We appreciate your critical insights and provide detailed clarifications below.
>
> Please refer to this **[Anonymized GitHub Link](https://anonymous.4open.science/r/RLPO-9577/Rqpq/readme.md)** where we have compiled detailed explanations for better understanding.
>
> ## Q1: Concerns about seed prompt acquisition.
>
> We agree with the reviewer’s observation that our current setup is primarily focused on low-level visual concepts. But our current setup can easily be modified to capture higher-level semantic concepts (e.g., gender, age) by adding task specific questions like, “What is the gender of the person in the image?” or “Which age category does the person in the image belong to (young/old)?”.
>
> To verify the usability of the proposed methodology with higher-level semantic concepts like gender, we trained a Resnet18 classifier on CelebA dataset to classify images as “Blonde” and “Not Blonde”. This dataset is known for having a spurious correlation between class “Blonde” and females. With our method, we were able to find the same correlation. Concepts  generated for the female face were more important than male face. As a side note, when we train RLPO to capture higher-level semantic concepts, it starts combining one or more than one low-level features. As shown in examples on **[anonymized github](https://anonymous.4open.science/r/RLPO-9577/Rqpq/readme.md)**, the generated samples start developing long and blonde hairs for both male and female concepts.
>
> ## Q2: Concerns regarding limited evaluation.
>
> We would like to clarify that, while the main text includes examples like “zebra” and “tiger” for clarity and interpretability, RLPO was evaluated on a broader range of randomly selected samples from ImageNet (see Section 4.3, Appendix D.3, and Table 4). RLPO has also been tested on multiple pretrained models such as GoogleNet, InceptionV3, indicating RLPO is not tied to a specific architecture.
>
> Additionally, as mentioned in Appendix D.9, rather than predominantly featuring specific cases, while conducting the human survey we considered 10 unique classes and randomly selected examples. Apart from that, we also demonstrate how RLPO generalized to non-visual domains like sentiment analysis (see Section 4.6 and Fig. 8).
>
> ## Q3: Justification for the assumption that $\text{C}_H \subseteq \text{C}_G$ and $\text{C}_R \subseteq \text{C}_G$.
>
> Intiutively, the generative models like Stable Diffusion are trained to learn the distribution of real-world data (which contains human defined and retrieved concepts). By leveraging this learned distribution, they can create existing or new data which by design can represent human defined or retrieved concepts (or neither). That being said, to solidify this assumption we plotted the clip embeddings of generated, retrieved and human defined stripes concept. For retrieval based concepts we used the stripes collected by CRAFT for the zebra class, for human defined concepts we used the concepts collected by TCAV authors for the zebra class, and for generated concepts we used pre-trained stable diffusion 1.5 to generate random stripes images. As shown in the plot on **[anonymized github](https://anonymous.4open.science/r/RLPO-9577/Rqpq/readme.md)**, the generated stripes encapsulates both human-defined and retrieved concepts hence proving the assumption $\text{C}_H \subseteq \text{C}_G$ and $\text{C}_R \subseteq \text{C}_G$.

---

> > ### Comment · Reviewer_Rqpq · 2025-04-06
> >
> > After a detailed review of the authors' rebuttals and other peer comments, I have decided to increase my rating to 3.
> >
> > I would like to highlight that I still have reservations regarding the generalizability of the proposed method. The limitations primarily stem from the seed prompts and biases inherent in the diffusion model being used. While the authors' rebuttal has addressed some of my concerns, not all have been fully resolved. For example, the case presented in the rebuttal depends on human knowledge to craft task-specific questions for improving seed prompt acquisition. Additionally, the justification for the underlying assumptions is based on examples from only a specific class ('zebra') within the dataset.
> >
> > Nevertheless, I appreciate the novel approach of employing RL and generative models to generate dataset-dependent concepts for use in concept-based explanation methods. To the best of my knowledge, this is the first and it addresses a significant challenge in concept-based explanation methods: identifying concept sets that effectively reflect model behaviors. I did not want to dismiss a contribution that leverages recent advances in generative models to tackle the challenge, as it shows promising potential, in view.
> >
> > I would suggest that the authors acknowledge these limitations in the final version of the paper, if it is accepted. Future work—by the authors or the broader XAI community—could further enhance the method’s generalizability by refining the seed prompting process, leveraging more advanced generative models, and exploring additional improvements.

---

> > > ### Author Response · Authors · 2025-04-08
> > >
> > > We thank the reviewer for their vote of confidence in our paper! Following the reviewer’s suggestion, we will include a discussion in the revised manuscript highlighting the reliance of our method on seed prompts and include the future work in that direction. We sincerely appreciate the reviewer’s constructive feedback and recognition of our method’s potential.

---

### Official Review · Reviewer_ADMa · 2025-03-13

**Overall Recommendation:** 3

**Summary:**

This work introduces an RL-based method to construct a vision-language concept-level preference dataset purely from synthesized images by taking TCAV score as the reward. It first prompts the trained MLLM to ground the common concepts represented as language phrases in the images. Then, it generate preference datasets with TCAV scores of ImageNet pre-trained models. Finally, the RLPO is applied to finetune the diffusion models. The results demonstrate the proposed method can guide generative models to generate more clustered concept images.

**Claims And Evidence:**

Are the claims made in the submission supported by clear and convincing evidence? If not, which claims are problematic and why?

I think it is overclaim on Understanding Neural Networks’ Internal Representations, since it does not propose novel explainable methods to unveil the concepts of learned vision networks.

Also refer to **Relation to Broader Scientific Literature.**

**Essential References Not Discussed:**

It seems to miss a line of related works on concept learning with diffusion models.

For example

ConceptLab: Creative Concept Generation using VLM-Guided Diffusion Prior Constraints

**Experimental Designs Or Analyses:**

See weaknesses.

**Methods And Evaluation Criteria:**

The proposed method or framework is technically novel.

**Other Comments Or Suggestions:**

I would suggest that the author polish this script entirely for another round.

**Other Strengths And Weaknesses:**

Weaknesses:

- Not well-organized: The current version of the presentation does not meet the ICML standards, therefore I would suggest the author revise their writing to clarify and reorganize the method and experimental sections. For instance, the evaluation metrics and motivations behind them are not clarified. Some expressions are not academic and rigorous enough.
- Missing technical details: to my understanding, the original C-deletion is performed in the pixel space, but this method is performed in the textual space.
- Potential semantic leakage in diffusion alignment since the vision models are already trained on ImageNet? We more details on the concept vocabulary, how many of semantic names are out of ImageNet vocabulary? How many classes can it generate? Can it generate novel categories/objects?
- The action space is not scaled, only a few words. Besides, most prompts are single phrases, and thus cannot scale up to diverse compositions?
- What if generated images in both groups are both grabage? Will it still update the SD? It is necessary to include additional quantitative metrics to distinguish between diversity and quality. It may need human study or auxiliary scores, e.g., train an additional network to compute classification accuracy.

**Questions For Authors:**

None

**Relation To Broader Scientific Literature:**

I am a little concerned about the implication of this work: this work essentially is attempting to align the vision-language alignment of generative models pre-trained with noisy image-text pairs. It cannot generalize beyond the abstract visual concepts that cannot be described by language. Therefore, the ultimate results achieved is more related to aligning the generative model in the word or phrase level.

**Theoretical Claims:**

None

---

> ### Author Rebuttal · Authors · 2025-04-01
>
> Please refer to this **[Anonymized GitHub Link](https://anonymous.4open.science/r/RLPO-9577/ADMa/readme.md)** where we have compiled detailed explanations for better understanding.
>
> We are thankful to the reviewer for the feedback. While we appreciate the reviewer's recognition of the proposed framework, we believe there are some key misunderstandings regarding the claims and the experiments conducted in the paper.
>
> In this work, we propose a novel approach to “generate” concepts that truly matter to the neural network by utilizing reinforcement learning-based preference optimization (RLPO) on diffusion models. And, we show qualitatively and quantitatively that the generated concepts matter to the neural network. This has been supported by Reviewer 1, 2 and 4. With regards to experiments, please see our detailed answers below.
>
> ## Q1: Generalizability beyond visual concepts.
>
> RLPO does not solely depend on direct linguistic grounding. Instead it uses reinforcement learning to evolve and refine the generative model via XAI feedback (from the model under test) - indirectly capturing internal representation that may not have exact language equivalents. The use of language (seed prompts), serves primarily to narrow the search space and provide a reasonable initialization for the concept generation. While the generated concepts may initially align with the seed prompts, RLPO iteratively optimizes the generation process via TCAV-based rewards, allowing for drift and refinement toward more model-relevant abstractions—even beyond the original prompt scope. Additionally, to highlight the generalizability of our approach beyond visual concepts, in Section 4.6 (Fig. 8), we demonstrate how RLPO generalized to non-visual domains like sentiment analysis, using textual input and preference.
>
> ## Q2: Clarification on C-deletion.
>
> We agree with the reviewer that the original C-deletion is performed in the pixel space. But we would like to clarify that, the C-deletion we show in the paper is also performed in the pixel space and **not in the textual space**. As highlighted in Q1, the use of language is to narrow down the search space. Once the training is completed, we generate concepts via the trained diffusion model and map those generated concepts back to the input space using CLIPSeg (as shown in Figure 5).
>
> The C-deletion graphs shown in paper are obtained from deleting most relevant to least relevant target concepts from the input images. We have provided more elaborate examples in Fig. 21 and 22 of Appendix.
>
> ## Q3: Potential semantic leakage in diffusion alignment since the vision models are already trained on ImageNet? We need more details on the concept vocabulary, how many semantic names are out of ImageNet vocabulary? How many classes can it generate? Can it generate novel categories/objects?
>
> We understand the reviewer’s concern on potential semantic leakage in diffusion alignment since these models have already been trained on ImageNet data, but we would disagree with the reviewer because the concepts we generate don't come from class data. As shown in Table 4, the generated concepts by RLPO are farthest from the class data. On a contrary, this semantic leakage occurs in retrieval based methods where the concepts are collected from class data. This is the exact problem we are trying to resolve with our method.
>
> ## Q4: The action space is not scaled, only a few words. Besides, most prompts are single phrases, and thus cannot scale up to diverse compositions?
>
> Our current action space consists of 20 seed prompts, preprocessed and extracted using VQA (see Appendix C.3). We would like to emphasize that our approach does not rely on a direct mapping from seed prompts to the final generated outputs (concepts). While the generated concepts may initially align with the seed prompts, RLPO iteratively optimizes the generation process via TCAV-based rewards, allowing for drift and refinement toward more model-relevant abstractions—even beyond the original prompt scope. Consequently, the final generated concepts capture more diverse and model-relevant abstractions that extend beyond the limitations of the initial single phrases.
>
> ## Q5: What if generated images in both groups are both garbage? Will it still update the SD?
>
> If both image sets yield low TCAV scores (i.e., do not activate the model meaningfully), our method does not update the diffusion model. Only when a concept has the potential to move toward explainable states do we update the diffusion model. We will clarify this further in the revised manuscript.

---

> > ### Comment · Reviewer_ADMa · 2025-04-09
> >
> > I appreciate the efforts made by the authors for the rebuttal. My questions have largely been addressed. For Q3 and Q4, I intended to ask about more results on the expanded vocabulary and evaluate with more words out of the ImageNet vocab as initial seeds. For Q5, I intended to ask for additional analysis on the failure patterns of behaviors of generative models in this loop.
> >
> > Raised score by one, considering some challenges of this work can be further extended in the future.

---

### Official Review · Reviewer_Sob6 · 2025-03-14

**Overall Recommendation:** 3

**Summary:**

The paper reframes the concept set creation as a concept generation problems. It proposes a method based on generative model to generate concept images. It aims to create to reliably generate diverse concepts that are challenging to craft manually. The process involves various components including Reinforcement Learning-based Preference Optimization, Stable diffusion models, Testing with Concept Activation Vectors (TCAV) scores, BLIP models.

## update after rebuttal
I increased my score to 3 after the rebuttal from the authors. They were actively engaged in addressing my concerns, and most of them have been resolved. However, I still have some reservations regarding the reliance on the seed prompt and the variability of the CAV. While the authors provided some results related to this issue, they are limited to only a few examples and do not fully address my concerns.

**Claims And Evidence:**

- The paper claims that RLPO generates new concepts that explain model behavior. However, it also acknowledges that results depend heavily on the text prompts used. If concept discovery relies significantly on the seed prompt, the method may not be truly generating new concepts but rather refining existing ones based on prior knowledge. The generated images appear to closely follow the provided seed prompts. The authors argue that explaining concepts through images is more intuitive, citing “A picture is worth a thousand words, but words flow easier than paint.” However, they do not fully justify why image-based explanations are inherently superior, especially given the reliance on text prompts for concept generation.
- The method relies on TCAV, which has known limitations. One key issue is the choice of hyperparameters, such as selecting which layers to focus on, which can significantly impact results. Additionally, unless many samples are used for training the classifier and obtaining Concept Activation Vectors (CAVs), the CAVs show a large variance across runs. The paper does not fully address these concerns.
- The method is highly complex, involving multiple components such as reinforcement learning, preference optimization, and generative modeling. This complexity itself is a drawback for model explanation methods, which should ideally be simple and interpretable. Additionally, RLPO depends on many pretrained models such as Stable Diffusion and BLIP, which introduces external biases that are not accounted for in the paper.

**Essential References Not Discussed:**

It would be helpful to discuss how this paper relates to the following works.
https://distill.pub/2017/feature-visualization/ Generating concepts images without any dependence on external input (such as BLIP's prompt seed suggestion in this paper)
https://arxiv.org/abs/2312.02974, https://arxiv.org/abs/2410.05217 While not involving classifiers, they also try to define abstract concepts without manual curation.

**Experimental Designs Or Analyses:**

- The paper does not discuss how the authors chose the hyperparameters for TCAV, particularly which layers were selected for extracting activations. This is important, as TCAV scores are known to vary significantly depending on this choice.

**Methods And Evaluation Criteria:**

The paper presents various evaluations, each assessing different aspects of the method, which is great. However, trade off between the dependence on seed prompts and the degree of genuine concept discovery may be further clarified.

**Other Comments Or Suggestions:**

No more comments.

**Other Strengths And Weaknesses:**

Please see above.

**Questions For Authors:**

- In Figure 6, it is unclear what x-axis “Steps” refers to.

**Relation To Broader Scientific Literature:**

The paper is addressing an important problem of concept generation. However, it is unclear to me how methodologically it would have advantage over existing methods.

**Theoretical Claims:**

No formal proofs were reviewed in detail. However, the effectiveness of RLPO is largely tied to TCAV, which is known to be sensitive to hyperparameter choices. The paper does not extensively analyze the robustness of TCAV in this setting.

---

> ### Author Rebuttal · Authors · 2025-04-01
>
> We appreciate the reviewer’s constructive comments and recognition of the importance of concept generation in our work. Our method improves upon traditional automatic concept retrieval approaches. While conventional methods extract concepts directly from the dataset—risking semantic information leakage—our approach generates novel concepts that are independent of the dataset.
>
> We have addressed the reviewers' concerns as follows.
>
> Please refer to this **[Anonymized GitHub Link](https://anonymous.4open.science/r/RLPO-9577/Sob6/readme.md)** where we have compiled detailed explanations for better understanding.
>
> ## Q1: Reliability on text prompts.
> Our use of seed prompts serves primarily to narrow the search space and provide a reasonable initialization for the concept generation. Given that Stable Diffusion is conditioned on text prompts, starting with semantically meaningful phrases helps the RL agent converge faster toward relevant concept regions.
>
> While the generated concepts may initially align with the seed prompts, RLPO iteratively optimizes the generation process via TCAV-based rewards, allowing for drift and refinement toward more model-relevant abstractions—even beyond the original prompt scope. We demonstrate this evolution through multiple RL steps (e.g., Figure 2, Appendix D.4), and also show in ablation (Appendix C.4) that random prompts perform significantly worse, indicating the importance of a good starting point rather than dependence.
>
> ## Q2: Hyperparameters used for TCAV score calculation.
> TCAV can be sensitive to choices like the layer of activation and the classifier used. To address this, as stated in Appendix C.1, we replaced the default SGD classifier with Logistic Regression, which we found provided more stable CAVs with lower variance. Regarding the choice of layers and target classes, we provide details in Appendix C.3.
>
> ## Q3: Bias introduced by pre-trained models.
> We acknowledge that the diversity of generated outputs depend on the generative model's capabilities. Issues such as insufficient representation of certain patterns could limit the range of explanations. However, such limitations are not unique to generative approaches—they are also inherent to retrieval-based methods, which are similarly constrained by available data and even human collected concepts, which are influenced by cognitive bias. However, we agree that this is a good point and we will include a discussion of this limitation in the revised manuscript. Specifically, we will highlight the dependency of the explanations on the generative model's capability to produce high-quality and diverse outputs. Thank you for pointing it out.
>
> ## Q4: In Figure 6, it is unclear what x-axis “Steps” refers to.
> “Steps” in figure 6 refers to a step in c-deletion. At each step we delete a part of the image representing the target concept. We have provided more elaborate examples in Fig 21 and 22 of Appendix. We will clarify this in the figure caption of the revised manuscript.
>
> ## Suggestions on related works
> Thank you for the suggested references. We will include a discussion of Feature Visualization (Olah et al., Distill) and the more recent diffusion-based concept learning works (e.g., https://arxiv.org/abs/2312.02974, https://arxiv.org/abs/2410.05217). While our method relies on classifier feedback via TCAV and thus differs in motivation, your point about manual curation versus automated discovery is well-taken and worth addressing more directly in Section 2.

---

> > ### Comment · Reviewer_Sob6 · 2025-04-06
> >
> > Thank you for the detailed response. However, I will maintain my score as my key concerns remain insufficiently addressed:
> >
> > The authors note that (1) the method performs poorly without seed prompts, and (2) the final generated concepts differ from the seed prompts, to justify the method's reliance on seed prompts. However, these points do not address the core concern—whether the method is truly capable of discovering new concepts, rather than merely refining or drifting from the initial prompt space.
> >
> > While the authors state that they used logistic regression instead of SGD to reduce the variance of CAV, I could not find clear evidence or quantitative analysis showing the effectiveness of the change in Appendix C.1 they referred to. An analysis demonstrating the extent to which this change mitigates the variance issue will be needed. Without such an analysis, it's difficult to assess the robustness of TCAV-based feedback in this context.

---

> > > ### Author Response · Authors · 2025-04-08
> > >
> > > Thank you for your response and for continuing to engage with our work! We have clarified the reviewer’s concerns on capability of the proposed method in comparison with discovery and refinement, and provided additional analysis on stability of logistic regression classifier while calculating CAVs. Since we have clarified all reviewer’s concerns, we sincerely hope the reviewer can reconsider the score.
> > >
> > > ## Q1: Clarification on Discovery vs. Refinement?
> > > We understand the reviewer’s concern regarding whether our method discovers new concepts or merely refines those provided via seed prompts. We would like to clarify that both processes are integral to our approach. Our method is a two-step loop:
> > > 1. **Discovery through RL** – The reinforcement learning (RL) agent searches for high-reward regions in the concept space by selecting from an initial set of seed prompts. Over time, the RL policy learns to favor prompts (or their refinements) that generate samples strongly aligned with the target class. As shown in Appendix D.3.1, cumulative rewards—computed using TCAV scores—increase consistently across classifier models, indicating that the RL agent is learning to select more informative prompts.
> > > 2. **Refinement through Diffusion Update** – For each selected prompt, we generate two concept sets, compute TCAV scores, and update the diffusion model to enhance alignment with the more salient concept set. This process introduces drift from the initial prompt toward more model-relevant abstractions. As illustrated in Appendix D.4 and Fig. 24, applying RLPO to the seed prompt “zoo” for the tiger class results in a progression of outputs—from general zoo-related imagery to increasingly tiger-specific features such as orange-black stripes and whiskers.
> > >
> > > Thus, while seed prompts serve as initial anchors, the joint RL and diffusion optimization progressively steers the concept generation process toward novel and more informative representations—extending well beyond the original prompt space.
> > >
> > > ## Q2: Stability of logistic regression classifier while calculating CAVs.
> > > We appreciate the reviewer’s request for empirical evidence regarding the stability of TCAV scores when using a logistic regression classifier instead of the original SGD-based one. To this end, we conducted an additional experiment using the “stripes” and “dots” concepts from the original TCAV paper, evaluating the "zebra" class on two different model architectures and activation layers. For each configuration, we computed TCAV scores across five independent runs, comparing the mean and standard deviation between the logistic regression (our implementation) and SGD (default implementation) classifiers.
> > >
> > > As shown in the table below, logistic regression results in low or no variance and provides more stable TCAV scores, validating its use in our setup. In response to the reviewer’s suggestion, we will include this experiment in the revised manuscript to better support our design choice.
> > >
> > > | Model     | Layer       | SGD - Stripes/Random      | Logistic - Stripes/Random | SGD - Dots/Random        | Logistic - Dots/Random     |
> > > |-----------|-------------|---------------------------|----------------------------|---------------------------|----------------------------|
> > > | GoogleNet | inception3a | 0.662 ± 0.03 / 0.338 ± 0.03 | 0.67 ± 0.00 / 0.33 ± 0.00  | 0.36 ± 0.05 / 0.64 ± 0.05 | 0.33 ± 0.00 / 0.67 ± 0.00  |
> > > |           | inception4e | 0.992 ± 0.01 / 0.008 ± 0.01 | 1.00 ± 0.00 / 0.00 ± 0.00  | 0.01 ± 0.007 / 0.99 ± 0.007| 0.00 ± 0.00 / 1.00 ± 0.00  |
> > > | ResNet50  | layer3      | 0.796 ± 0.02 / 0.204 ± 0.02 | 0.78 ± 0.00 / 0.22 ± 0.00  | 0.078 ± 0.07 / 0.922 ± 0.07| 0.00 ± 0.00 / 1.00 ± 0.00  |
> > > |           | layer4      | 1.000 ± 0.00 / 0.000 ± 0.00 | 1.00 ± 0.00 / 0.00 ± 0.00  | 0.60 ± 0.54 / 0.40 ± 0.54 | 0.00 ± 0.00 / 1.00 ± 0.00  |

---

### Official Review · Reviewer_zFMD · 2025-03-16

**Overall Recommendation:** 4

**Summary:**

The authors proposed a method to discover and visualize the "concept" or hidden knowledge learnt by a neural network. They proposed a reinforcement learning framework to achieve this goal. A score (TCAV) was proposed to evaluate whether a hidden representation of NN forms a concept.

**Claims And Evidence:**

The authors provided quantitative and instance evidences that their algorithm successfully discovered hidden concepts.

**Essential References Not Discussed:**

To the best of my knowledge the references are appropriate.

**Experimental Designs Or Analyses:**

The experiments are valid.

**Methods And Evaluation Criteria:**

The evaluation metrics sounds rational.

**Other Comments Or Suggestions:**

N/A

**Other Strengths And Weaknesses:**

The work is of high novelty and would potentially impact a broad scope of machine learning. It would be better if more rigorous quantitative evaluations can be developed in the future (e.g., correlation between generated concepts vs. ground truth concepts). Also, the procedure of generating concept seeds are still a black-box (i.e., diffusion model), which should be improved in the future.

**Questions For Authors:**

1）It seems that the users need to have a good understanding of the target concept in order to generate good concept seeds (i.e, proper VQA design and possibly selective prompts). Is it possible to extract concepts from a large paragraph of description texts without prior human knowledge?
2）After the concept pictures are generated by SD+LORA, it is unsure how can you align them with the original input picture?

**Relation To Broader Scientific Literature:**

How machine learning models learn concepts of the world from data is of wide interest in establishing foundation of AI. This work is a very interesting attempt.

**Theoretical Claims:**

Theoretical analyses are provided. It is hard to follow the details but the framework sounds rational.

---

> ### Author Rebuttal · Authors · 2025-04-01
>
> Thank you for identifying the novelty and appreciating the experiments. We hope the following explanations will clarify the queries for the reviewer.
>
> ## Q1: It seems that the users need to have a good understanding of the target concept in order to generate good concept seeds (i.e, proper VQA design and possibly selective prompts). Is it possible to extract concepts from a large paragraph of description texts without prior human knowledge?
>
> We agree with the reviewer’s observation that in our current setup, in order to generate good concept seeds proper VQA design and selective prompts are needed. But because of the modularity of our approach, the generation of seed prompts can be replaced by any other concept generation method. As described in [1], the end user can extract concepts from text descriptions without prior human knowledge and directly use it in our proposed method.
>
> [1] Zang, Yuan, et al. "Pre-trained vision-language models learn discoverable visual concepts." arXiv preprint arXiv:2404.12652 (2024).
>
> ## Q2: After the concept pictures are generated by SD+LORA, it is unsure how you can align them with the original input picture?
>
> We employ CLIPSeg, a transformer-based segmentation model, to establish visual correspondence between generated concepts and regions in the input images (See Section 4.4 Figure 5). By feeding the generated concept images as prompts into CLIPSeg, we produce heat maps highlighting areas in the input image that resemble the generated concept. This allows us to localize abstract concepts (e.g., “stripes”, “mud”) within the original images, enabling interpretable alignment between concept space and class-specific features.
>
> We hope these responses clarify your concerns. We appreciate your recognition of the work's broader implications and suggestions for future extensions, which we plan to incorporate.

---

### Decision · Program_Chairs · 2025-05-01

**Decision:**

Accept (poster)

**Comment:**

This paper got 1 accept and 3 weak accept rating.

Before rebuttal,

Reviewers thought the strength of this paper are: 1) The work is of high novelty and would potentially impact a broad scope of machine learning (Reviewer zFMD); 2) The paper is addressing an important problem of concept generation (Reviewer zFMD); 3) The proposed method or framework is technically novel (Reviewer ADMa); weaknesses are: 1) problems of the seed prompt (Reviewer zFMD, Sob6, Rqpq); 2) The method relies on TCAV, which has known limitations (Reviewer Sob6); 3) The method is highly complex, involving multiple components (Reviewer Sob6); 4) paper is Not well-organized (Reviewer ADMa); 5) Missing technical details (Reviewer ADMa); 6) The action space is not scaled, only a few words (Reviewer ADMa); 6) hold reservations about the qualitative results of the methods in broader cases (Reviewer ADMa).

After rebuttal,

All the reviewers confirmed they had read authors' response and would like to update the review if needed. Reviewer zFMD kept the rating. Reviewer Sob6 raise the score, but still had some reservations regarding the reliance on the seed prompt and the variability of the CAV and thought authors' did not fully addressed their concerns. Reviewer ADMa also raised score. Reviewer Rqpq raised score but still had reservations regarding the generalizability of the proposed method.

Considering this AC decided to give weak accept rating and hope authors could incorporate reviewers' comment to improve the paper.